# Phenolic Phytochemicals for Prevention and Treatment of Colorectal Cancer: A Critical Evaluation of In Vivo Studies

**DOI:** 10.3390/cancers15030993

**Published:** 2023-02-03

**Authors:** Samhita De, Sourav Paul, Anirban Manna, Chirantan Majumder, Koustav Pal, Nicolette Casarcia, Arijit Mondal, Sabyasachi Banerjee, Vinod Kumar Nelson, Suvranil Ghosh, Joyita Hazra, Ashish Bhattacharjee, Subhash Chandra Mandal, Mahadeb Pal, Anupam Bishayee

**Affiliations:** 1Division of Molecular Medicine, Bose Institute, Kolkata 700 054, India; 2Department of Biotechnology, National Institute of Technology, Durgapur 713 209, India; 3Jawaharlal Institute Post Graduate Medical Education and Research, Puducherry 605 006, India; 4College of Osteopathic Medicine, Lake Erie College of Osteopathic Medicine, Bradenton, FL 34211, USA; 5Department of Pharmaceutical Chemistry, M.R. College of Pharmaceutical Sciences and Research, Balisha 743 234, India; 6Department of Pharmaceutical Chemistry, Gupta College of Technological Sciences, Asansol 713 301, India; 7Department of Pharmacology, Raghavendra Institute of Pharmaceutical Education and Research, Anantapur 515 721, India; 8Department of Biotechnology, Indian Institute of Technology, Chennai 600 036, India; 9Department of Pharmaceutical Technology, Jadavpur University, Kolkata 700 032, India

**Keywords:** colorectal cancer, phenolic compounds, prevention, treatment, molecular mechanisms, in vivo

## Abstract

**Simple Summary:**

Colorectal cancer (CRC) is a significant cause of death worldwide. The inefficacy of the current treatment regimens is reflected in the frequent recurrence and emergence of a drug-resistant form of CRC. Numerous published reports from independent investigators around the globe have shown the great potential of natural products as a source of anti-CRC drug-leads with novel functions. Here, we have reviewed the literature on phenolic phytochemicals carrying anti-CRC activity in various in vivo models and analyzed their molecular basis of action to understand the implications of these findings in the future treatment and prevention of CRC.

**Abstract:**

Colorectal cancer (CRC) is the third most diagnosed and second leading cause of cancer-related death worldwide. Limitations with existing treatment regimens have demanded the search for better treatment options. Different phytochemicals with promising anti-CRC activities have been reported, with the molecular mechanism of actions still emerging. This review aims to summarize recent progress on the study of natural phenolic compounds in ameliorating CRC using in vivo models. This review followed the guidelines of the Preferred Reporting Items for Systematic Reporting and Meta-Analysis. Information on the relevant topic was gathered by searching the PubMed, Scopus, ScienceDirect, and Web of Science databases using keywords, such as “colorectal cancer” AND “phenolic compounds”, “colorectal cancer” AND “polyphenol”, “colorectal cancer” AND “phenolic acids”, “colorectal cancer” AND “flavonoids”, “colorectal cancer” AND “stilbene”, and “colorectal cancer” AND “lignan” from the reputed peer-reviewed journals published over the last 20 years. Publications that incorporated in vivo experimental designs and produced statistically significant results were considered for this review. Many of these polyphenols demonstrate anti-CRC activities by inhibiting key cellular factors. This inhibition has been demonstrated by antiapoptotic effects, antiproliferative effects, or by upregulating factors responsible for cell cycle arrest or cell death in various in vivo CRC models. Numerous studies from independent laboratories have highlighted different plant phenolic compounds for their anti-CRC activities. While promising anti-CRC activity in many of these agents has created interest in this area, in-depth mechanistic and well-designed clinical studies are needed to support the therapeutic use of these compounds for the prevention and treatment of CRC.

## 1. Introduction

The diagnosis of colorectal cancer (CRC) is a death sentence to many. CRC is the third most diagnosed and second leading cause of cancer mortality worldwide [1]. In the United States alone, there were 149,500 new cases and 52,980 deaths in 2021, with an estimated 151,030 new cases for 2022 [1]. Globally, there were 1.9 million new cases and 935,000 deaths in 2020 [2]. These numbers have risen since 2018, as at that time statistics were noted to be 1.8 million new cases and 861,000 deaths [3]. Analyses predicted the global CRC burden to rise by 60% to 2.2 million new cases and 1.1 million deaths by 2030 [3,4,5,6]. Rising cases are attributed to a more sedentary lifestyle and altered dietary habits, such as consuming processed foods, tobacco usage, and heavy alcohol consumption. India’s incidence of colon cancer in 2016 was estimated to be 63,000, with a sizeable interstate variation [7,8].

Since the implementation of a screening program in the United States in 1990, CRC incidence has consistently decreased in the population of those older than 50 years [9,10]. In contrast, CRC incidence has shown a significant and steady increase (2% per year) in the population of those less than 50 years of age, which is called young-onset CRC (yCRC) [9,11,12]. While yCRC comprises only 10% of total CRC incidence, 75% of yCRC incidence affects the population of those between 40 and 49 years of age [9,11,12,13,14,15]. A study undertaken between 1975 and 2010 predicted that yCRC would double by 2030 in the U.S. population of those younger than 35, indicating racial disparity [9,11,12,13,14,15].

Current treatment options available for colorectal cancer include laparoscopic surgery, resection, palliative, neoadjuvant chemotherapy, and radiotherapy [15,16,17,18,19,20,21,22]. Chemotherapy causes undesirable side effects. In addition to being frequently ineffective, current treatments are expensive.

Utilizing phytochemicals for cancer treatment and prevention has been a matter of serious discussion for decades [3,23]. Plants have been used to treat many diseases in traditional medicine and have been a forefront in alternative approach. Over 3000 plant species have anticancer activities, with thirty plant-derived compounds undergoing preclinical testing [5]. Anticancer activity in citrus fruits, allium vegetables, and medicinal plants has demonstrated preclinical success [5,8]. Secondary plant metabolites have been shown to decrease inflammation and increase apoptosis in addition to possessing antioxidant, anticarcinogenic, and antimetastatic properties [8,23,24]. The attraction to phytochemicals arises from relatively safer and cost-efficient natural products, and their consumption by humans is widespread [5]. While research is being conducted, often with promising results, only a limited number of natural compounds have been approved for clinical use, while the clinical application of many is hindered due to low bioavailability [5,23].

Numerous literature reviews and studies on natural compounds in CRC were dissected and sorted thoroughly for relevant and vital information. It was noted that very few articles reviewed CRC and the therapeutic prospects with polyphenols [25,26]. There is no review literature explaining all classes of phenolic compounds and their signaling pathways in contrast with CRC. We have also noted that few previous reviews have focused on using plant extracts and fractions rich in phenols and pure phenolic compounds [25,26]. Some have examined flavonoids and their effects on CRC [27,28,29,30,31,32,33,34,35,36], yet no such reviews consider other classes of phenolic compounds and their effects on CRC. In contrast, numerous reviews were dedicated to discussing the deadly disease of CRC, but did not examine natural products for its treatment. A few reviews that included CRC studied general nutrition and dietary effects, but the literature examined dietary products, such as calcium, fiber, processed meats, or medicinal plants, rather than plant phenolic compounds [37,38,39,40,41]. Furthermore, a review was noted to include the effects of phytochemicals on CRC, but only mentioned specific biochemical properties and pathways of cancer development [42]. In view of the aforementioned limitations, our present review is up-to-date and offers the most recent information compared to previously published works. In this review, we first evaluated pertinent literature to present the characteristics of CRC and identify common risk factors and current treatment options. Then, we evaluated various in vivo studies on different phenolic phytochemicals to understand the potential of these natural agents for CRC prevention and treatment. We hope these phenolic phytocompounds spark interest in conducting new studies to eventually aid in decreasing the prevalence and lowering the risk of CRC.

## 2. Risk Factors

Familial, hereditary, and lifestyle factors are independent risk factors for developing CRC [43]. Genetic syndromes comprise 20–30% of CRC cases and can be divided into non-polyposis and polyposis types (Table 1). Lynch syndrome, an alternate term for the non-polyposis syndrome, is an autosomal dominant disease associated with a defect in DNA mismatch repair genes, such as *hMLH1*, *hMSH2*, *hMSH6*, or *hPMS2* [44,45]. This mutation results in microsatellite instability (MSI) regions, which is also associated with ~15% of sporadic CRC cases. As expected, individuals with MSI regions carry an increased risk for other cancers, such as endometrial carcinoma [44].

Familial adenomatous polyposis syndrome (FAP), which is characterized by multiple polyp formations in the gastrointestinal tract, is caused by a germline mutation in the adenomatous polyposis coli (APC) gene [67,68,69]. Inheriting a polyposis syndrome can increase an individual’s risk of developing colon cancer up to 100% [70]. Furthermore, these patients carry the risk of developing other gastrointestinal cancers and desmoid tumors. MUTYH-associated polyposis (MAP), Peutz-Jeughers syndrome (*STK11*), Juvenile polyposis syndrome (*SMAD4* and *BMPR1A*), hyperplastic polyposis (HPP), familial CRC (FCC) syndrome X, and Cowden syndrome (*PTEN*) are other polyposis syndromes that predispose individuals to an increased risk of developing CRC [50,71,72].

Chronic inflammatory bowel diseases, which encompass both ulcerative colitis and Crohn’s disease, predispose individuals to CRC [73]. Additionally, previous abdominopelvic radiation is a potent risk factor for CRC, especially for childhood cancer survivors [74]. Furthermore, individuals receiving prostate cancer-related radiation therapy are at a higher risk of developing rectal carcinoma, supporting previous radiation therapy as a risk factor for CRC [75]. Cystic fibrosis is also implicated in CRC, as there is a 5–10 times greater risk of acquiring CRC in these patients. As a result, they have a separate management for CRC screening, especially post-transplant [76].

Lifestyle patterns, such as smoking, consumption of alcohol, obesity, sedentary lifestyles, and chronic diseases, pose a potent overall risk of developing sporadic CRC [77,78,79]. A westernized diet, rich in processed foods and red meat and deficient in fruits, fiber, and leafy vegetables, can contribute to CRC development [16,80]. Conversely, consuming more vegetables, fruits, and fiber is protective against CRC. A meta-analysis has elucidated the risk of CRC with food’s dietary inflammatory index (DII). A higher DII correlating with a pro-inflammatory state increases CRC risk [81]. Numerous studies have explored the opposite end of the spectrum, examining anti-inflammatory foods and drugs for CRC chemoprevention and treatment. This is supported by a case-control meta-analysis where a higher intake of calcium, magnesium, and potassium lowered the occurrence of CRC [82].

The risk of CRC is low in vegetarians compared to meat eaters with an HR ratio of 0.49 [95% confidence interval (CI): 0.36 to 0.66], and 0.73 [95% CI: 0.54 to 0.99] when not adjusted and adjusted (for sociodemographic and lifestyle factors, multimorbidity, and body mass index) respectively. When CRC was subcategorized, the HR of 0.69 [95% CI: 0.48 to 0.99] for the colon and 0.43 [95% CI: 0.22 to 0.82] for the proximal colon was observed in vegetarians, which is much less compared to meat eaters [83]. Adherence to the Mediterranean diet was found to be associated with a low risk of rectal cancer with RR of 0.82 [95% CI: 0.71 to 0.95] for rectal cancer, 0.94 [95% CI: 0.87 to 1.02] for proximal colon cancer, and 0.91 [95% CI: 0.79 to 1.04] for distal colon cancer [84]. The unhealthy diet pattern is associated with CRC-specific mortality with RR/HR of 1.52 [95% CI: 1.13 to 2.06] [85]. The high intake of dietary calcium and magnesium is negatively associated with CRC risk with HR of 0.76 [95% CI: 0.72 to 0.80] and 0.80 [95% CI: 0.73 to 0.87], respectively. The higher intake of dietary heme, however, was positively correlated to colon cancer incidence with HR of 1.01 (95% CI: 0.82 to 1.19) and rectal cancer incidence with HR of 1.04 [95% CI: 0.67 to 1.42] [82]. The increase in DII score, and CRC are found to be positively associated with an overall increased risk of CRC by 40% with RR of 1.40 [95% CI: 1.26 to 1.55] [81]. Smoking and CRC shows a positive association with ever smoker versus never smoker, the pooled RR was 1.18 [95% CI: 1.11 to 1.25], and the pooled risk estimate was 1.25 [95% CI: 1.14 to 1.37] [77]. Alcohol consumption is also associated with an increased risk for CRC mortality. In comparison, the pooled RR was 1.03 [95% CI: 0.93 to 1.15] for any, 0.97 for light drinkers who consume ≤12.5 g of ethanol/day, 1.04 [95% CI: 0.94 to 1.16] for moderate drinkers who consume 12.6–49.9 g ethanol/day), 1.04 [95% CI: 0.94 to 1.16] for heavy drinking men (who consume ≥50 g ethanol/day), which is higher than heavy drinking women [pooled RR = 0.79 (95% CI: 0.40 to 1.54)] [78].

## 3. Pathogenesis

Overall, the pathogenesis of colon cancer involves three main pathways: the chromosomal instability (CIN)/classic adenoma-carcinoma sequence [86,87], the CpG island methylator phenotype (CIMP), and the microsatellite instability (MSI) pathway [88]. While these are separate pathways, there is potential overlap within them. Moreover, they involve the stepwise accumulation of multiple mutations, eventually leading to CRC development [89].

The classic adenoma-carcinoma sequence accounts for 65–70% of sporadic diseases commonly observed as left-sided CRCs [90]. This mechanism involves a dysfunctional/inactivated APC gene located on chromosome 5q21. APC, a “gatekeeper” of colonic neoplasia, has been implicated in familial adenomatous polyposis (FAP) syndrome. The onset of CRC is inevitable in a population with an inactivating mutation in both copies of the APC gene [91,92]. APC controls cell growth and differentiation through the Wnt/β-catenin signaling pathway. The Wnt pathway is an essential cellular signaling system by which several developmental events for embryological and tissue homeostasis occur, involving cellular proliferation and differentiation. Deregulation of the Wnt pathway can lead to the development of cancer. When the Wnt/β-catenin pathway is suppressed, there is a lower rate of cellular proliferation and fewer intestinal stem cells [93]. Activating mutations of Wnt/β-catenin leads to the pathogenesis of CRC. Over 90% of CRC cases carry mutations within this pathway [94]. It has been found that APC deletion/loss of function leads to CRC development, while restoring APC function can regress adenomas by reducing Wnt activity [93].

Apart from APC, there are other Wnt activating mutations, such as mutations in the CTNNB1 gene encoding β-catenin. R-spondins are another module of Wnt signal activators, which are associated with up to 10% of CRC mutations. Antagonism of RSPO3 with paclitaxel effectively targeted Wnt signaling in CRC [95]. A higher expression of ß-catenin in CRC cells is associated with a worse prognosis and advanced stage of the disease. Because of this, CRC metastasis was determined by the combined β-catenin odds ratio in the nucleus [96].

In the absence of APC function, β-catenin translocate to the nucleus. In cooperation with the DNA binding factor TCF, it promotes the growth of colonic epithelium via uncontrolled overexpression of its targets c-Myc and cyclin D1 [93]. Next, a mutation in KRAS contributes to molecular pathogenesis by promoting adenoma formation [97]. Finally, a mutation in p53 facilitates the progression of CRC [98]. Although important roles of p53 and KRAS were implied in the adenoma-carcinoma pathway, mouse knockout of APC develops carcinoma irrespective of its KRAS and p53 status, and re-introduction of APC restores cellular differentiation and normal crypt formation [43,93].

The microsatellite instability pathway occurs due to the inactivation of DNA mismatch repair genes, which includes ATPases hMSH2, hMSH6, hMSH3, hMLH1, hPMS2, hPMS1, and hMLH3, as involved in Lynch syndrome [99]. The MSI pathway is involved in roughly 15% of CRCs, 3% of which are Lynch syndrome while the rest are sporadic, mainly caused by MLH1 hypermethylation. Finally, the CpG island methylator phenotype (CIMP) is involved in silencing genes by hypermethylation of CpG islands on their promoters [100,101]. CIMP has been associated with older patients, female patients, and right-sided lesions with high MSI and BRAF mutations. CIMP is also associated with PI3K mutations but lacks KRAS and p53 mutations. A clearer insight and greater understanding of CIMP is required to better study the treatment and prevention of CRC [102].

## 4. Chemoprevention

Chemoprevention aims to intervene, prevent, suppress, and reverse the initiation and progression of carcinogenesis. It further attempts to decrease the recurrence of cancer through the usage of drugs, vitamins, and nutritional supplements [66]. Various agents, including nonsteroidal anti-inflammatory drugs (NSAIDs), such as aspirin, and other agents, such as metformin, statins, minerals, and vitamins, have been previously studied for their chemopreventive benefits regarding CRC (Table 2). There is little doubt that a significant stride has been made into the unventured territories for the chemoprevention of CRC.

In CRC involving the APC/β-catenin pathway, cyclooxygenase-2 (COX-2) is often implicated in the early and later stages of the adenoma sequence, driving the formation into a carcinoma [120,121,122,123]. Furthermore, COX-2 overexpression produces vascular endothelial growth factor (VEGF), which promotes tumor angiogenesis [124,125]. Hence, by targeting COX-2, various studies have shown that NSAIDs, ranging from aspirin and sulindac to the more selective COX-2 inhibitors, such as celecoxib, have proven benefits in reducing disease risk [126,127]. In the 1990s, the U.S. Preventive Services Task Force recommended aspirin to prevent non-high-risk CRC [128,129,130].

Other drugs, such as metformin, showed promising effects in reducing the risk of CRC development. Recent meta-analyses showed that metformin could reduce CRC risk by 22% [131]. In an ongoing ASAMET trial for the tertiary prevention of stage I–III CRC, patients were administered low doses of aspirin combined with metformin for a potential synergistic chemo-preventive action [132]. Statins, a specific inhibitor of HMG-CoA reductase in the mevalonate synthesis pathway, have been recommended to lower serum lipid levels [133]. Statins were shown to reduce CRC alone and in combination with NSAIDS [134,135]. Further investigations on multiple agents, such as antioxidants, minerals, such as selenium, and vitamins, including A, C, E, and β-carotene, were previously believed to have benefits in decreasing the risk of CRC, yet they have yielded mixed results [130,136,137]. Studies on folate’s use to lower CRC risks also yielded mixed results [130]. Fiber, alcohol, monounsaturated fatty acids, polyunsaturated fatty acids, omega-3, omega-6, niacin, thiamine, riboflavin, vitamin B6, vitamin B12, zinc, magnesium, selenium, vitamin A, vitamin C, vitamin D, vitamin E, folic acid, β-carotene, anthocyanin, flavonoids, garlic, ginger, onions, thyme, oregano, saffron, turmeric, rosemary, eugenol, caffeine, and tea have all demonstrated anti-inflammatory benefits, and therefore reduce the risk of CRC development [138,139]. A higher intake of dietary fiber, pertaining to whole grains, was associated with a lower CRC risk in men [140].

## 5. Treatment

CRC incidence and mortality have been efficiently controlled by the routine screening and removal of polyps through colonoscopy [141]. Surgery, chemotherapy, and immunotherapy are mainstay treatments for CRC; the stage of CRC progression in each patient determines an appropriate combination. The treatment of CRC depends upon the diagnosis through tumor/node/metastasis (TNM) staging of the lesion. Adjuvant chemotherapy with fluorouracil (5-FU) decreases death rates in patients with high-risk stage II colon cancer by 3–5% and 10–15% in stage III disease alone [142]. MSI/MMR protein levels determined by IHC aid in deciding the adjuvant therapy [143,144,145]. Furthermore, after primary tumor resection, TNM or immunoscore can be considered to assess the tumor recurrence risk [146].

Single-agent therapy with 5-FU or therapy with multiple agents composed of 5-FU and oxaliplatin (FOLFOX), 5-FU and irinotecan (FOXFIRI) (IRI), or capecitabine and oxaliplatin (CAPOX), capecitabine (CAP), and irinotecan (CAPIRI) as first line chemotherapy is recommended based on the sensitivity and the stage of the disease. In many cases, single-agent chemotherapies yielded better results than combination therapy, given the associated systemic toxicity and unsatisfying responses [147,148,149]. A combination of 5-FU or CAP with oxaliplatin (OX) is recommended for stage III CRC for three to six months. Patients with intermediate-risk stage II CRC are recommended either 5-FU or CAP, which are added to OX, if the patients are high risk (stage II), for three months [145]. The International Duration Evaluation of Adjuvant Chemotherapy (IDEA) collaboration helped investigate whether three or six months of adjuvant chemotherapy was necessary, as cumulative toxicity develops from fluoropyrimidines/oxaliplatin in the form of peripheral sensory neuropathy. Results show that the overall disease-free survival was similar at 74.6% and 75.5% for three months and six months, respectively. After three months of treatment, the overall sensory peripheral neuropathy reduced from 34% to 11%. However, per ESMO guidelines, stage III CRC should still be treated with six months of FOLFOX or CAPOX if the patient falls within the high-risk category. For patients who do not tolerate oxaliplatin, capecitabine, or LVGFU2 can be acceptable alternatives [145]. 

Various forms of supplemental targeted immunotherapies are considered to aid chemotherapy. Monoclonal antibodies are used to attack various potential genes, such as ERFR, VEGF, and PDL-1/PDL-1. Cetuximab, an anti-EGFR chimeric monoclonal antibody, and bevacizumab, an anti-VEGF chimeric monoclonal antibody, both of which prolong OS, were the first line targeted drugs approved by the United States Food and Drug Administration (FDA) in 2004 [150,151]. An immune checkpoint blocker α-PD1/PDL-1 antibody, in combination with chemo- and radiation therapy, was approved by the FDA for MSI-H and dMMR classes of CRCs for sustained progression-free survival [152]. Cetuximab yielded a positive outcome for CRC that did not respond to single-agent IRI or fluoropyrimidine therapy. Combining cetuximab with IRI, fluorocytidine, or OX delivered promising results [151,153]. EGFR (epidermal growth factor receptor) is overexpressed in various cancers to different extents, including 25–75% in CRC [154]. Cetuximab, once bound, results in the internalization and degradation of EGFR [111]. However, cetuximab was inactive in CRCs carrying the RAS (KRAS) mutation. Like EGFR, the VEGF level is also elevated in CRC, predicting a poor prognosis [155]. Along with an elevated VEGF level, increased vascular endothelial growth factor receptor (VEGFR) activity is found in adenomas, as well as in the metastatic stage of CRC [147,156]. While cetuximab is not suitable as a second line agent, bevacizumab is often an excellent choice.

## 6. Literature Search Methodology

We have followed the Preferred Reporting Items for Systematic Reviews and Meta-Analysis (PRISMA) guidelines [157] for this work. Four scholarly databases, namely PubMed, Scopus, ScienceDirect, and Web of Science, were utilized to screen the literature for the last 20 years (2002 to 2022 November) by searching the title, abstract, and key words section with the key words “colorectal cancer” AND “phenolic compounds”, “colorectal cancer” AND “polyphenol”, “colorectal cancer” AND “phenolic acids”, “colorectal cancer” AND “flavonoids”, “colorectal cancer” AND “stilbene”, and “colorectal cancer” AND “lignan.” All search results were gathered, and duplicate files were removed. Next, literature was scanned based on title and abstract. Selected articles were then searched for retrieval. After reading the full articles, preclinical studies (in vivo animal models) with polyphenols were selected and incorporated. The methodology for literature search and study selection is depicted in Figure 1.

## 7. Phenolic Compounds with In Vivo Anti-CRC Activities

Plants synthesize phenolic compounds as secondary metabolites and carry multiple aromatic rings with two or more hydroxyl groups. Phenolic compounds carry a wide (~8000 different) variety of chemical structures. Based on chemical structures, phenolic compounds are divided into different classes, such as flavonoids (e.g., anthocyanidins, flavanols, flavanones, flavones, flavonols, and isoflavoniods) and non-flavonoids, including phenolic acids (e.g., hydroxycinnamic acids and hydroxybenzoic acids), coumarins, stilbenes, lignans, and tannins [158,159,160]. Significant sources of phenolic compounds are fruits and vegetables. Various phenolic compounds are known for their interesting pharmacological properties, including antioxidant, anti-inflammatory, neuroprotective, and anticancer properties [161,162].

While western medicines have significant effects on CRC chemoprevention and treatment, extracts of numerous plants and plant products are still currently in use, as humanity has used plants for thousands of years as traditional or ethnic medicines for the prevention and treatment of various ailments, including cancer. The primary reasons for their popularity are fewer side effects, easy availability, and low cost compared to synthetic drugs [163,164,165]. Over the last several decades, steady progress has been achieved in identifying the bioactive secondary metabolites of plants, such as phenolic compounds, and understanding their mode of action to explain their health benefits [166,167,168,169]. In the following sections, we aim to summarize the anti-CRC effects of phenolic compounds based on animal studies. Table 3 describes the in vivo CRC activity of the compounds as revealed by our literature search as depicted in Figure 1. We have selected 16 relatively well-studied compounds to describe their anti-CRC activities in a greater detail in the following sections. The chemical structures of various classes of phenolic compounds with in vivo anti-CRC activities are presented in Figure 2, Figure 3, Figure 4 and Figure 5. 

### 7.1. Flavonoids

#### 7.1.1. Baicalin

Baicalin (molecular weight: 446.4 g/mol), conjointly called baicalein 7-O-glucuronide and 7-D-glucuronic acid-5, 6-dihydroxyflavone or known by its chemical name, 5, 6 dihydroxy-4-oxo-2phenyl-chromen-7-yl) oxy-3, 4, 5-trihydeoxytetrahydropyran-2-carboxylic acid, is a glycosyloxyflavone. It is a key component of a variety of traditional medicine preparations, consisting of Sho-Saiko-To, Yangkun pills, Kushen decoction, and Shuanghuanglian injections. Scutellariae radix, Scutellaria planipes, Scutellaria rehderiana, and Scutellaria scandens are only a few of the Scutellaria species that contain the compound baicalin, which is extensively distributed throughout the genus [358].

Baicalein suppressed AOM/DSS-induced colon tumors in mice and induced apoptotic cell death. Baicalein suppressed inflammation by PARP1-mediated NF-κB inhibition [180]. A dose of 50 mg/kg baicalin suppressed the growth of highly metastatic SW620 tumor xenograft in BALB/c nude mice [181]. Baicalin inhibited the TLR4/NF-κB signaling and significantly suppressed CT-26 tumor growth, migration, and invasion. Anti-tumor immunity was also enhanced by an increase in CD4^+^ and CD8^+^ T cells in CT-26 tumors [182]. Baicalein treatment induced apoptosis in a p53-mediated Akt-dependent manner and suppressed HT-29 tumor xenograft [183]. In another study, baicalein suppressed MMP-2 and MMP-9 and inhibited DLD1 tumor growth and metastatic effects by inhibiting phosphorylation of ERK [184]. 

Dou et al. [185] showed that baicalein and baicalin can significantly inhibit the growth of HCT116 tumor xenograft by inducing tumor cell apoptosis and senescence through inhibiting the telomerase reverse transcriptase. It has also been hypothesized that the control of colon cancer cell apoptosis and senescence is caused by the MAPK, ERK, and p38 signaling pathways. Wang et al. [186] verified that baicalin application increased the expression of DEPP and triggered its downstream target Ras/Raf/MEK/ERK and p16INK4A/Rb pathways by serving as an antioxidant, resulting in senescence in colon carcinoma cells in HCT116 tumor model in BALB/c athymic nude mice. It was revealed that baicalin inhibited the HT-29 xenograft tumor in nude mice by suppressing c-Myc as the driver of miRNAs responsible for oncogenic development (oncomiRs). These findings demonstrated an association of c-Myc in baicalin-mediated inhibition of colon cancer growth [187]. In orthotopically transplanted tumors of CRC cells in BALB/c nude mice, baicalin administration lowered the levels of marker proteins for cell cycle, EMT, and stemness [188].

Wang et al. [189] observed that the baicalein therapy dramatically decreased tumor numbers in the small intestine and colon of Apc^Min/+^ mice. Furthermore, reduced levels of inflammatory cytokines, such IL-1, IL-2, IL-6, G-CSF, and GM-CSF B, in this mouse tumor model suggested that baicalein’s anti-CRC action was mediated by reducing gut inflammation. Baicalin treatment suppressed HCT116 tumor xenograft growth by downregulation of CircMYH9 and HDGF, and upregulation of miR-761 [190].

#### 7.1.2. Curcumin

Curcumin, with the chemical name (1E, 6E)-1,7-bis(4-hydroxy-3-methoxyphenyl)-1,6-heptadiene-3,5-dione(diferuloylmethane), is a hydrophobic polyphenol derived from the roots of a well-known Indian spice, turmeric (Curcuma longa). Consumption of turmeric is believed to provide protection from numerous ailments, including CRC [359,360,361,362]. Anti-CRC activities of curcumin were demonstrated by several independent groups. Curcumin reduced putative precursor colonic lesions, e.g., aberrant crypt foci (ACF), through suppressing the levels of proinflammatory cytokines, such as TNF-α and IL-6, and proinflammatory mediators, such as COX-2, in obese and diabetic (db/db) mice [197]. Adiponectin plays an important anti-inflammatory role in the gut [363,364,365]. Curcumin increased the adiponectin level in both AOM-treated and untreated C57BL/KsJ-db/db (db/db) mice [197]. Leptin levels are directly proportional to body fat. High serum leptin levels can cause inflammation-mediated CRC [366,367]. Curcumin was able to reduce the body fat content along with serum leptin levels, and thus reduce the severity of CRC. This study also observed AMPK activation and COX-2 inhibition in those animals [197].

Curcumin reduced DSS-induced ACF and β-catenin accumulation. Due to its anti-inflammatory properties, curcumin suppressed pro-inflammatory cytokines and COX-2 and iNOS in DSS-induced colonic tissue [194]. Curcumin suppressed the growth of HCT116 tumor xenograft in ICR SCID mice. Curcumin treatment led to proteasome inhibition and induction of apoptosis which, in turn, suppressed the HCT116 tumor growth [195]. In another study, curcumin inhibited AOM/DSS-induced tumorigenesis in mice. Curcumin also downregulated Axin2 and exerted its anticancer activity by Axin2 mediated inhibition of the Wnt/β-catenin pathway [196].

Curcumin was found to inhibit HCT116-induced xenografts in male nude mice, along with suppressing NF-κB regulated genes, including Bcl-2, c-FLIP, IAP1, and survivin. It further cleaved procaspase-3 and procaspase-9. Curcumin pretreatment sensitized the tumor xenograft to γ-radiation and suppressed NF-κB activity by inhibiting the binding of NF-κB to its response element on its target genes, thus minimizing invasion, migration, and angiogenesis. Curcumin ameliorated the γ-radiation mediated increase of cellular proinflammatory mediator COX-2 and c-Myc in a HCT116 xenograft tumor model [198,199]. 

Furthermore, curcumin was found to modulate gut microbiome habitat in AOM-injected IL10-/-mice and was implicated in the function of anti-inflammation and the maintenance of gut homeostasis. The aberrant cytoplasmic and nuclear localization of β-catenin in AOM-treated wild-type and AOM/Il-10-/-mice was significantly reduced by curcumin treatment [200].

Curcumin enhanced the anti-CRC activity of capecitabine in HCT116 tumor xenografts in male athymic nu/nu mice through the induction of apoptosis and inhibition of angiogenesis, invasion, and metastatic factors, such as VEGF, ICAM-1, and MMP-9, and CXCR4. Inhibition of COX-2 and cell cycle progression mediators, cyclin D1 and c-Myc, was also observed in the curcumin-treated animals. The anti-CRC effects of liposomal curcumin alone and combined with oxaliplatin were tested on CRC xenografts induced by Colo205 and LoVo cells in athymic nu/nu mice. The combination therapy showed efficient tumor growth inhibition by apoptosis (PARP-1 cleavage). Liposomal curcumin also inhibited angiogenesis in consistence with the inhibition of VEGF, CD31, and IL-8 expression [201]. Phytosomal curcumin was tested for its ameliorative effects on an AOM/DSS model of colitis-associated CRC alone and in combination with 5-FU in in vivo. Curcumin, alone and in combination, functioned through modulating Wnt/β-catenin signaling and E-cadherin activities. Curcumin administered by oral gavage and in combination with 5-FU significantly inhibited GSK3 α/β and cyclin D1 expression. Curcumin was shown to reduce oxidative stress induced ACF and colon injuries induced by AOM/DSS by upregulating endogenous antioxidative enzymes, such as superoxide dismutase (SOD), catalase (CAT), thiolase, and inducing autophagy by upregulating beclin1 [200]. 

#### 7.1.3. Catechins

Catechins are a group of polyphenols abundantly present in tea, cocoa, berries, grapes, and apples. Catechins have a myriad of health benefits, and their anticancer properties have been extensively studied [368,369]. Kim et al. [370] examined the effects of green tea polyphenol (GTP) dosage on DSS-induced acute colitis and DMH and DSS-induced colon cancer developed in male ICR mice. GTP contained 70% of total catechins, half of which were EGCG and 3% being caffeine. This study showed that a specific dosage of GTP was effective in ameliorating the carcinogenic effect of DSS/DMH. The basis of this activity was implicated in the antioxidant properties of GTP. If the dosage was higher or lower than the effective dose, GTP was ineffective. This is potentially due to a loss of, or insufficient, antioxidant properties. Depending on the treatment conditions, GTFP can exhibit antioxidant or pro-oxidant properties [371].

The anticancer effect of EGCG was also tested on azoxymethane (AOM)-induced male C57BL/KsJ-db/db (db/db) mice. EGCG caused a significant reduction in the levels of IGF-IR, phospho-IGF-IR, phospho-GSK-3β, β-catenin, COX-2, and cyclin D1. There was also an increase in serum IGFBP3 and a decrease in serum IGF-I, insulin, triglyceride, cholesterol, and leptin in the treated mice [206].

Zhong et al. [207] investigated the acetylated-EGCG activity against protumorigenic inflammatory mediators in AOM-mediated colitis-induced CRC in a male mouse model. Acetylated-EGCG inhibited the expression of pro-tumorigenic inflammatory mediators, such as inducible nitric oxide synthase (iNOS) and COX-2. iNOS is one of the enzymes that remain in ACF and causes the continuous formation of nitric oxide (NO), leading to the promotion of tumorigenesis [372,373,374]. Furthermore, COX-2 converts arachidonate to prostaglandin E2. A sustained overexpression of prostaglandin E2 in the tissues may lead to epithelial cell cancers, including CRC [207,375,376].

EGCG showed the antistemness and chemosensitizing effects on xenograft tumors of HCT116 spheroid-derived cancer stem cells in male nude mice. EGCG inhibited CRC stemness and sensitized 5-FU-resistant HCT116 cells. EGCG suppressed stemness markers, such as Notch-1, and upregulated the expression of tumor suppressive miRNAs, including miR34a, miR200c, and miR-145 [208].

Another study demonstrated the effects of green tea catechins alone and in combination with curcumin on DMH-induced colon cancer in male Wistar rats [209]. A 32-week-long dietary treatment with curcumin, green tea catechins, and their combination showed a significant reduction in the number of colorectal aberrant cryptic foci in these animals. Notably, the combinatorial treatment had a greater effect than that with either of the compounds acting alone. A significant decrease in the proliferation index and an increase in the apoptotic index were reported in the groups treated with a combination of the compounds, compared to the mock-treated group or those receiving only DMH [209].

The anticancer effect of polyphenol E (PPE) was tested on AOM-treated F344 rats. PPE is a standardized GTP mixture containing 65% EGCG and other catechins. After AOM treatment, the animals were given a 20% high-fat diet, with or without 0.24% PPE for 34 weeks. PPE treatment resulted in a significant reduction in tumor size and the number of tumors in these animals. PPE was shown to decrease nuclear β-catenin levels, induce apoptosis, and increase the levels of RXR-α, RXR-β and RXR-γ expression in adenocarcinomas. This was accompanied by the lowering of proinflammatory eicosanoids, prostaglandin E2, and leukotriene B4 in the plasma [276].

#### 7.1.4. Fisetin

Fisetin is a hydroxy flavone under the subgroup of flavonoid found in various fruits and vegetables, such as strawberry, apple, persimmon, grapes, onion, and cucumber. In an AOM/DSS-induced colitis associated CRC model in BALB/c mice, fisetin suppressed dysplastic lesions through inducing apoptosis in the colonic tissue along with downregulation of Bcl-2 and STAT3, and upregulation of cleaved-caspase-3 and BAX. Fisetin treatment restored the level of enzymatic (SOD, CAT, GPx, and GR) and non-enzymatic (vitamin E, and vitamin C) antioxidants in DMH-induced colonic tissue back to normal [213].

Fisetin treatment resulted in activation of AMPKα and inhibition of PI3K/Akt/mTOR signaling pathway along with decreased expression of PI3K, reduced Akt phosphorylation in PIK3CA mutants. In FC^1^3K^1^Apc^Min/+^ mice, fisetin decreased the occurrence of colonic tumor incidences. In combination with 5-FU, fisetin reduced the overall colonic tumor incidences [214].

Fisetin inhibited growth of LoVo tumor xenograft in athymic nude mouse model. Mechanistic study revealed that fisetin acted by inducing apoptosis in tumor tissue through activation of caspase-8 and increased cyt. c expression. In the tumor tissue of treated animals, inhibition of IGF1R and Akt activation was observed [215].

Although CT-26 tumor growth was suppressed upon the intratumoral injection of fisetin, HCT116 tumors were not sensitive to the similar treatment where a combination of radiation with fisetin was more effective. Fisetin suppressed the oncoprotein securin in CT-26 tumor in a p53-independent fashion, but securin null HCT116 tumors are more sensitive to fisetin treatment [216].

Fisetin suppressed HCT116 induced tumor growth in NOD/Shi-scid-IL2R gamma (null) (NOG) mice in a dose-dependent manner compared to control group [218]. Another study showed that due to poor water solubility, the fisetin micelles, composed of poly(ethylene glycol)-poly(ε-caprolactone), i.e., MPEG-PCL, are more efficient antitumor agents over free fisetin as tested in CT-26 tumor model. MPEG-PCL showed enhanced inhibition of angiogenesis through inducing apoptotic cell death [217].

#### 7.1.5. Genistein

Genistein, a naturally occurring isoflavone, was first isolated from Genista tinctoria. Its anticancer properties have been extensively studied [377]. Sekar et al. [222] examined genistein’s role in regulating the tumor microenvironment in DMH-induced colon cancer in Wistar rats. This study revealed that genistein could regulate enzymatic (SOD, CAT, GPx, and GR) and non-enzymatic (vitamin E, vitamin C, vitamin A, and GSH) antioxidants in DMH-induced colonic tissue environments. It was found that the loss of mucin secretion in DMH-induced animals was restored by genistein. There was also a reduction of mast cell population and collagen deposition in genistein-treated animals compared to mock-treated animals. Argyrophilic nuclear organizer region and proliferating cell nuclear antigen, two prognostic markers, were decreased by genistein in DMH-treated rats. Genistein activated NRF2 and its downstream target, heme oxygenase-1, and alleviated DMH-induced oxidative stress. Higher expression of colonic stem cell markers, such as CD133, CD44, and β-catenin, was found to be reduced by genistein in DMH-treated animals [222].

It was shown that oral administration of genistein to mice carrying orthotopically implanted human CRC did not inhibit tumor growth. However, it did show inhibition of distant metastasis formation at a dose non-toxic to the animals. Subsequent biochemical analyses showed genistein-mediated downregulation of matrix metalloproteinase-2 (MMP-2) and FMS-related tyrosine kinase 4, also known as vascular endothelial growth factor receptor 3, suggesting its inhibitory role against neoangiogenesis in mouse tumors [224].

Chen et al. [378] conducted a study in which clinical signatures of the anti-CRC activity of genistein were tested in clinical samples of plasma, tumor tissue samples, and standard tissue samples isolated from patients. The expression of miR-95, serum glucocorticoid kinase 1 (SGK1), Bcl-2, and Erk1 was highly elevated in samples of CRC compared to the normal samples. Furthermore, genistein could sensitize CRC SW620 cells to apoptosis with increased LDH content in a concentration-dependent manner, accompanied by downregulation of endogenous miR-95, SGK1, and Erk1 activities [378].

Zhang et al. [223] studied the effect of genistein on AOM-induced colon carcinogenesis in male Sprague Dawley rats. The animals were given a control diet, soya protein isolate (SPI), and a genistein diet orally, starting from gestation to 13 weeks of age. Pre-AOM treatment analysis was performed by taking samples at seven weeks of age, and the remaining rats were AOM-treated at this time for six weeks for analysis. Compared to the control group, AOM injections did not cause aberrant nuclear accumulation of β-catenin in SPI and genistein-treated groups. Moreover, SPI and genistein suppressed the expression of cyclin-D1 and c-Myc. In addition, the expression of Wnt signaling genes (Wnt5a, Sfrp1, Sfrp2, Sfrp5) was decreased to a level comparable to that of pre-AOM treatment by SPI and genistein. Furthermore, the rats fed SPI and genistein had lower numbers of total aberrant crypts, which correlated with the reduction in Wnt/β-catenin signaling. Genistein also lowered the number of ACF [223].

The first clinical study to assess the safety and tolerability of genistein in combination with chemotherapy for the treatment of metastatic CRC was conducted by Pivota et al. [379]. Patients diagnosed with metastatic CRC but not previously treated were administered FOLFOX or FOLFOX-bevacizumab. Genistein (60 mg/day) was given orally for seven days every two weeks. Treatment was started four days before chemotherapy and continued through days one through three of infusion chemotherapy. In this trial, thirteen patients received combinatorial treatment. Treatment with genistein alone resulted in mild side effects, such as headaches, nausea, and hot flashes, with one subject experiencing grade 3 hypertension. There was no increase in chemotherapy-related adverse events when genistein was added to FOLFLOX. The best overall response rate for the genistein supplementation of the chemotherapy regimen was 61.5%. The median progression-free survival of the same study was 11.5 months [379].

#### 7.1.6. Kaempferol

Kaempferol, a dietary flavanol found in many plants, including apple, tea, broccoli, and grapefruit, has been demonstrated to carry antitumor effects based on preclinical studies [380]. Nirmala et al. [239] demonstrated the beneficial effects of orally administered kaempferol with intravenous irinotecan in 1,2-dimethyl hydrazine (DMH)-induced colorectal carcinoma in male Wistar rats. In the kaempferol-fed animal groups, levels of DMH-induced erythrocyte lysate levels and decreased liver thiobarbituric acid reactive substances. Levels of several antioxidant enzymes, such as catalase, superoxide dismutase, and glutathione peroxidase, were recovered, and the most successful effects were achieved at a dose of 200 mg/kg body weight of kaempferol (which is comparable to irinotecan). 

The combined effect of fluoxetine, an antidepressant drug, and kaempferol in alleviation of DMH-induced colon carcinoma in male Sprague Dawley rats was also analyzed. Compared to fluoxetine and kaempferol alone, combined treatment of these two agents caused greater reduction in multiple plaque lesions and preneoplastic lesions in the colonic tissues. This combinatorial treatment also reduced tissue concentration of malondialdehyde and NO. Both serum and tissue β-catenin levels were significantly decreased by the combinatorial treatment. There was also a significant increase in serum and tissue caspase-3 levels. PCNA and COX-2 positive cells in the colon of animals receiving the combinatorial treatment were lower when compared to fluoxetine and kaempferol treatments alone [240]. 

Hassanein et al. [241] studied the effect of sulindac in combination with either EGCG or kaempferol in DMH-induced colon carcinogenesis in male Sprague Dawley rats. The combinations of EGCG and kaempferol with sulindac, a nonsteroidal anti-inflammatory drug, caused great enhancement of sulindac’s antioxidant, anti-inflammatory, antiproliferative, and apoptotic activities. Sulindac combined with both compounds caused a decrease in thiobarbituric acid-reactive substance, tissue NO, and both serum and tissue β-catenin. Downregulation of PCNA and COX-2 and a decrease in the number of ACF caused by DMH administration were also noted [241]. 

#### 7.1.7. Luteolin

Luteolin (3′,4′,5,7-tetrahydroxyflavone) was discovered in different fruits, vegetables, and medicinal herbs. Plants rich in luteolin are used for treating various ailments, such as hypertension, inflammation, and cancer in Chinese traditional medicine [381,382]. The anti-CRC activity, as well as the anti-angiogenic, anti-invasive, and antimetastatic effects of luteolin were studied using AOM-induced colitis models of male BALB/c mice. Upregulation of γ-glutamyl transferase (GGT), found in a number of human neoplasms, facilitates neoplastic progression and metastasis [246,383]. GGT and 5′-nucleotidase (5′ND) were inhibited in AOM-treated mice by luteolin. Furthermore, luteolin reduced other tumor markers in AOM-treated animals, such as cathepsin-D and carcinoembryonic antigen (CEA), which are correlated with poor prognosis [246]. Luteolin inhibited invasion and metastasis by reducing the expression of MMP-2 and MMP-9 along with enhancing expression of tissue inhibitor metalloproteinases 2 (TIMP-2) [246]. Mast cells were associated with enhanced angiogenesis and tumor malignancy [384]. It was found that luteolin also decreased giant mast cell and total mast cell populations in AOM-treated mice, compared to AOM-induced control animals [246].

Luteolin reduced the number and size polyps of DSS-treated mice. Upon luteolin treatment, DSS-induced oxidative stress, level of carcinoembryonic antigen and COX-2 were decreased in colonic tissue [242]. In another study, luteolin was shown to suppress AOM-induced CRC by downregulating iNOS and COX-2 expression level [243]. Luteolin also suppressed AOM-induced CRC by activating Nrf2/Keap1 pathway [244]. 

Luteolin inhibited HT29 xenograft’s growth in nude mice by an activity consistent with modulation of miR-384/pleiotrophin axis [247]. miR384 expression was found to be downregulated in the majority (83%) of CRC biopsy samples, correlating with the invasiveness and migratory abilities of CRC [385]. Pleiotrophin plays a major role in angiogenesis through upregulation of VEGF in CRC [386]. Luteolin treatment of HT-29 cell-induced xenograft tumor developed in female nude BALB/c mice efficiently suppressed the migration of CRC cells from the spleen to the liver and metastasis through upregulation of miR-384/pleiotrophin axis. Luteolin upregulated the expression of miR-384, which, by targeting pleiotrophin expression, inhibited the expression of MMP-2, MMP-3, MMP-9, MMP-16, as well as invasion and metastasis of CRC [247]. Luteolin, in synergy with adenovirus CD55-TRAIL, inhibited HT-29 xenografts in female BALB/c nude mice through increasing the apoptotic activity [248]. 

In another study, luteolin showed antimetastatic activity against CT-26 lung metastasis by downregulating MMP-2 and MMP-9. MEK and Akt phosphorylation was suppressed by the inhibition of Raf and PI3K by luteolin [245].

#### 7.1.8. Myricetin

Myricetin (3,3′,4′,5,5′,7-hexahydroxyflavone), a naturally occurring flavonoid pigment, is typically present in fruits, herbs, and nuts. The presence of three hydroxyl groups at 3-′, 4-′, and 5′-carbon positions makes myricetin unique from other flavanols [387]. Studies by Nirmala and Ramachandran [257] showed the efficacy of myricetin on 1,2-dimethylhydrazine-induced rat colon carcinogenesis. They demonstrated that myricetin administration reduced the incidence of tumor-bearing rats and tumors in total. Furthermore, myricetin supplementation dramatically decreased intestinal tumorigenesis developed in adenomatous polyposis coli multiple intestinal neoplasia (APC^Min/+^) mice. Additionally, myricetin treatment improved the antioxidant enzymes, including catalase, glutathione peroxidase, and GSH, in a dose-dependent manner [257].

Li et al. [258] assessed the effectiveness of myricetin against intestinal tumorigenesis in adenomatous polyposis coli multiple intestinal neoplasia (APC^Min/+^) mice. Promoting apoptosis in adenomatous polyps, myricetin-fed APC^Min/+^ mice grew fewer, smaller polyps and did not appear to experience negative side effects. By modifying the GSK-3 and Wnt/-catenin pathways, lowering the levels of the proinflammatory cytokines IL-6 and PGE2, and downregulating the phosphorylated p38 MAPK/Akt/mTOR signaling pathway, myricetin prevents the growth of intestinal tumors [258].

AOM/DSS-induced mice were used by Zhang et al. [259] to test myricetin’s effectiveness against chronic colonic inflammation and inflammation-driven carcinogenesis. Myricetin significantly decreased the levels of inflammatory factors, such as TNF-, IL-1, IL-6, NF-B, p-NF-B, COX-2, PCNA, and cyclin D1, to inhibit the development of colorectal tumors and shrink colorectal polyps [259].

M10, a new derivative of myricetin, was tested by Wang et al. [205] to show that M10 inhibits robust endoplasmic reticulum (ER) stress-induced autophagy in inflamed colonic mucosal cells of AOM/DSS-induced mice model. The decreased levels of proinflammatory mediators, such CSF/M-CSF, IL-6, and TNF-α, in colonic mucosa and the prevention of the NF-κB/IL-6/STAT3 pathway, were shown to be associated with the antitumor activity [260].

#### 7.1.9. Naringenin

Naringenin, a flavonoid found mostly in citrus fruits and vegetables with no taste or color, carries antioxidant, anti-inflammatory, antiviral, antimicrobial, and antitumor properties [388]. In addition, naringenin was found to reduce the number of high multiplicity aberrant crypt foci (HMACF) by 51% and the proliferative index by 32% in an AOM-induced rat model. Here, naringenin was implied to prevent CRC through decreasing proliferation and increasing apoptosis of luminal surface colonocytes [261]. 

Naringenin inhibited a dextran sulfate sodium (DSS)-induced murine colitis model. The inhibitory action was correlated with the inhibition of iNOS, ICAM-1, MCP-1, COX-2, TNF-α, and IL-6 transcript levels. The decrease in TNF-α and IL-6 levels was consistent with the suppression of TLR4 mRNA and protein in the colon mucosa. LPS-induced nuclear translocation of p65/RelA was also inhibited by naringenin in RAW264.7 cells, suggesting its action through TLR4 inhibition [262].

6-C-(E-phenylethenyl)-naringenin (6CEPN) inhibited anchorage independent growth of CRC cells, as well as in a CRC-induced xenograft in a dose-dependent manner through the inhibition of COX-1, an underlying cause of malignant character of CRC cells [263]. 

Naringin was shown to reduce tumor size and growth of AMO or DSS-induced CRC model in C57BL/6 mice by suppressing ER stress-induced autophagy in colorectal mucosal cells [265]. Another study showed naringin-mediated inhibition of tumor cell proliferation and AOM-induced CRC through inducing apoptosis in an AOM-injected Sprague-Dawley rat model [266,389].

#### 7.1.10. Quercetin

Quercetin (3,4,5,7-pentahydroxyflavone), a polyphenolic flavonoid, was isolated from vegetables, fruits, grain, seeds, and tea [282,390]. Quercetin was shown to carry various pharmacological properties, including anticancer properties. It was further found to be effective against AOM/DSS-mediated colitis induced CRC and showed a decrease in mucin-depleted foci and aberrant crypt foci development [391]. In addition, quercetin treatment was shown to efficiently reduce AOM/DSS-induced inflammation, a major cause of colon carcinogenesis [282,392,393]. In another study, quercetin was found to restore leukocyte levels lost by AOM/DSS treatment. It was also noted that quercetin efficiently downregulated various oxidative stress-related markers, such as lipid peroxide (LPO), NO, SOD, glucose-6-phosphate (G6PD), and glutathione (GSH), explaining its role in neutralizing inflammation. The metabolic profiling of sera demonstrated the effect of quercetin through the downregulation of biomarkers that are upregulated in AOM/DSS-treated mice [282].

In a metastatic cancer model induced in BALB/c mice by CT-26 cells, quercetin was shown to be effective through inducing the intrinsic pathway of apoptosis, along with upregulating the p-38 MAPK pathway. Notably, quercetin function was correlated with modulation of the EMT markers, such as downregulation of N-cadherin, snail, MMP-2, and MMP-9, while E-cadherin, TIMP-1, and TIMP-2 were upregulated [283]. 

Quercetin augmented radio-sensitization of CRC cells observed in HT-29 tumor xenografts through induction of apoptosis. Combining quercetin with a low dosage of 5Gy radiation effectively suppressed CRC cell proliferation with little toxicity towards normal colonic epithelial cells, CCD-18Co. The combinational therapy was found to target cancer stem cells, as suggested by the reduction of cancer stemness factors, such as DCLK-1, CD24, Lgr5, CD29, and CD44, and the colonosphere formation. The proportion of CD133+ cells also decreased in DLD-1 and HT-29 cells under combinatorial treatments [284].

Li et al. [284] further observed that the combinational therapy of ionizing radiation and quercetin targets the notch-signaling pathway through the downregulation of γ-secretase. The combinational therapy of ionizing radiation and quercetin effectively reduced the expression of γ-secretase complex components nicastrin, PEN2, APH1, presenilin-1, and presenilin-2, which suppressed notch cleavage and thus notch signaling. The combination therapy also inhibited the expression of Jagged-1 and cleaved Notch-1 protein levels [284].

Quercetin induced antiproliferative activity and proapoptotic effects are mediated by the upregulation of cannabinoid receptor-1 (CB1-R) in AOM-treated mice. The downregulation of STAT3 and pSTAT3 was also observed [279].

When radiation therapy was used with quercetin treatment, it suppressed the tumor size of the DLD1 tumor xenograft in athymic nude mice, indicating that quercetin enhanced the radiosensitivity of DLD1 tumors [280].

#### 7.1.11. Rutin

Rutin, a glycosidic derivative of quercetin, is also known as quercetin-3-O-rutinoside or vitamin-P. It is known to carry antimicrobial, antifungal, anti-inflammatory, anticancer, and antiallergic properties, with poor solubility in water [394]. Rutin naturally occurs in various plants, including buckwheat, Mez, Labisia pumila, *Sophora japonica* L., Schum, *Canna indica* L., and *Ruta graveolens* L. [395,396]. In a dose-dependent manner, rutin suppressed SW480 cell-induced tumor growth in a tumor xenograft model without affecting the organ or body weight. In the same model, rutin was shown to enhance mean survival time by 50 days and suppressed angiogenesis through decreasing the serum VEGF level [285].

#### 7.1.12. Tangeretin

Fruits and vegetables contain a wide variety of flavonoids. Citrus fruit flavonoids exhibit various biological effects, such as anticancer and antitumor properties. For example, tangeretin, a polymethoxylated (5,6,7,8,4′-pentamethoxyflavone) flavone, is predominant in the peel of citrus fruits and is thought to operate as a natural resistance factor against pathogenic fungus. In addition, tangeretin has been demonstrated to have several biological properties, including the capacity to suppress cancer cell growth [397].

Bao et al. [291] sought to create a nano-system that included tangeretin (TAGE) and atorvastatin (ATST) and was embellished with RGD (cyclized arginine-glycine-aspartic acid sequences) to treat colon cancer. To assess the anticancer effects of these two drugs on colon cancer cells and in female BALB/c mice harboring cancer models, these researchers produced ATST and TAGE combination nanosystems; RGD-ATST/TAGE CNPs. Results indicated that the RGD-decorated nano-system was more hazardous to HT-29 cells than the undecorated nano-system and that the weight ratio of ATST to TAGE, at which the highest synergism was seen, was 1:1. The integrated nano-systems had a high in vivo biodistribution in the tumor site and effectively reduced in vivo tumor development without significantly harming the treated mice’s primary organs and tissues [291].

#### 7.1.13. Wogonin

The medicinal plant *Scutellaria baicalensis* and the traditional Chinese medicine of Huang-Qin (Scutellaria radix) include a significant active monoflavonoid called wogonin (5,7-dihydroxy-8-methoxyflavone). Wogonin has many therapeutic possibilities, including anti-inflammatory and anticancer effects. It has also been observed to inhibit the development of several types of cancer cells with excellent specificity between normal cells and cancer cells [398,399]. 

To study wogonin’s role in colitis-associated colorectal cancer (CAC), Yao et al. [298] developed the AOM/DSS-induced C57BL/6 mice paradigm. They discovered that wogonin markedly reduced the prevalence of tumors and prevented the growth of colorectal adenomas by lowering the expression and secretion of IL-6 and IL-1β, as well as decreasing the cell proliferation and expression of NF-κB in adenomas and adjacent tissues. Further, it increased Nrf2 nuclear translocation in those same tissues [298].

Feng et al. [299] evaluated wogonin’s anti-colon cancer effect in an AOM-DSS-induced CRC animal model. They discovered that wogonin decreased tumor abundance and kept colon length within normal range without adversely affecting other organs. In addition, wogonin administration inhibited the SW480 cell-induced xenograft growth in BALB/c mice. Another study, by You et al. [300], further examined the effects of wogonin in mice with colon cancer. Treatment with wogonin abrogated the survival and metastasis properties of colon cancer cells in vivo. A detailed analysis revealed that wogonin-mediated upregulation of p-YAP1 level was responsible for the observed anti-colon cancer effect. This suggested the involvement of the Hippo signaling pathway in the process.

### 7.2. Phenolic Acids

#### 7.2.1. Caffeic Acid

Caffeic acid (3,4-dihydroxycinnamic acid) is a nonflavonoid catechol with potent antioxidant properties. It is found in almost all plants as an intermediate in the lignin biosynthesis pathway. The prime source of caffeic acid is coffee. Caffeic acid possesses various pharmacological properties, such as antioxidant, anti-inflammatory, anticancer, and neuroprotective effects [400]. Caffeic acid, by direct interaction, inhibited MEK1 and TOPK activity in an ATP non-competitive manner. Kang et al. [303] conducted experiments using caffeic acid on a mouse tumor model. It demonstrated action by inhibition of ERK and p90RSK activation. Caffeic acid suppressed the TPA-induced activation of AP1, NF-κB, and ERK signaling, and thus neoplastic transformation induced by TPA, EGF, and H-Ras. Through inhibition of ERK functions, caffeic acid inhibited lung metastasis of CT-26 cells. This study also indicated the usefulness of caffeic acid in reducing ERK activity in patient tumor samples.

Caffeic acid effectively inhibited cancer stem cells (CSC) and reduced radiation-induced sphere formation of CD133+ and CD44+ CSC in two patient-derived tumor xenograft (PDTX) models of human CRC in immune-suppressed mice. In vivo, the radiation-induced elevation of PI3K/Akt signaling pathway was also suppressed by caffeic acid. In caffeic acid-treated xenograft samples, the abundance of CD133+ and CD44+ subpopulations of CSC cells were decreased. In addition, CD44+ and CD133+ cells of CRC lost their ability for self-renewal, migration, and CSC-like properties due to caffeic acid in a PDTX xenograft model. Inhibition of PI3K/Akt signaling was described as a significant mode of action caffeic acid in inhibiting CSC proliferation [304].

Both caffeic acid phenethyl ester (CAPE) and caffeic acid phenylpropyl ester (CAPPE) could inhibit HCT116 cell-induced tumor xenograft in immune-compromised mice through inhibition of PI3K/Akt and inactivation of mTORC1 by AMPK activation. Treatment with CAPE and CAPPE reduced the MMP-9 level at a non-hepatotoxic concentration. In addition, CAPE and CAPPE suppressed expression of cyclin D1, Cdk4, cyclin E, c-Myc, and N-cadherin, and upregulated p21 in vivo. Expression of tumor biomarkers, such as PCNA and FASN, was also suppressed by CAPE and CAPPE in tumor tissue [305].

CAPE and caffeic acid p-nitro-phenethyl ester (CAPE-pNO2) upregulated the levels of p53, p27, p21, cytochrome c (cyt. C), and cleaved caspase-3, but downregulated procaspase-3, Cdk2, and c-Myc in HT-29 tumor xenograft in mice. There was a dose-dependent inhibition of tumor growth and VEGF expression by these compounds, with no visible toxicity to normal cells [306].

Consumption of decyl caffeic acid inhibited tumor growth in mice with a HCT116-induced tumor xenograft. The mechanism of action involved the induction of cell cycle arrest at the S phase as well as autophagy [307].

#### 7.2.2. Gallic Acid 

Gallic acid (3,4,5-trihydoxy benzoic acid) is a naturally occurring polyhydroxy phenolic acid found as an active compound in various fruits, nuts, food compounds, vegetables, and numerous plants, such as green chicory, grapes, blackberries, raspberries, blueberries, and strawberries. Gallic acid is well known for its antimicrobial, antioxidant, anti-inflammatory, and anticancer potential [401,402]. In a dose-dependent manner, gallic acid was shown to inhibit DSS-induced colitis in mice through the inhibition of STAT3 phosphorylation [320]. This inhibitory mechanism includes reduced proinflammatory mediators Th1, TNF-α, and IL-6, and chemokines, such as KC and MCP-1 [320].

In another study, the inhibitory effects of gallic acid were tested in HCT116 and HT29 cells and tumor xenografts in BALB/c mice. The function of pro-oncogenic factors, such as Src, STAT3, EGFR, and Akt, along with key players in the apoptosis pathway were analyzed. The results demonstrated inhibition of STAT3 and Akt by inhibiting Src and EGFR functions. Furthermore, net enhancement of the cleaved caspase-3 and caspase-9 suggested the involvement of apoptosis as the mechanism behind cell death [321].

Gallic acid was shown to ameliorate ulcerative colitis-associated CRC induced in rats by TNBS treatment by modulating ferroptosis, an iron-dependent process of cellular necrosis [322]. Gene expression profiling interactive analysis (GEPIA) and bioinformatics analysis identified significant involvement of ferroptosis-related genes in CRC prognosis. This analysis indicated that eight ferroptosis-related genes are involved in cell survival. This docking study suggested that gallic acid could induce ferroptosis by modulating some of these genes [322].

### 7.3. Stilbenes

#### Resveratrol

Resveratrol (3,5,4′-trihydroxystilbene), a stilbenoid that can be found in peanuts, skin of red grapes, and blueberries, has been studied for its potential anticancer properties [403,404]. Saud et al. [350] used a mouse model with a knocked-out APC locus, and Kras activated specifically in the distant colon to study the effect of resveratrol on sporadic CRC. The mice received a diet supplemented with resveratrol (150 or 300 ppm) before the appearance of tumors. This resulted in a 60% inhibition of tumor production and loss of Kras expression in 40% of mice that developed tumors. Oral administration of resveratrol for tumor bearing mice resulted in complete tumor remission in 33% of mice and a decrease in tumor size in 97% of the remaining mice. Upregulation of miR-96, a negative regulator of Kras expression, in non-tumoral and tumoral colonic tissues suggested that resveratrol exerted its anti-CRC effects by downregulating Kras expression [350]. Alfaras et al. [351] examined the effects of oral administration of trans-resveratrol on DMH-induced precancerous colonic lesions in male Sprague-Dawley rats. This resulted in the reduction of aberrant cryptic foci by 52% and mucin depleted foci by 45% in the colon. In colonic contents, dihydroresveratrol was the most abundant compound detected, followed by trans-resveratrol and its derivatives [351]. Synergistic effects of resveratrol and curcumin on CRC were studied by Majumdar et al. [352]. 

One study analyzed the effects of resveratrol and its PLGA-chitosan based nanoformulation in animal models (both xenograft and orthotopic) of colon cancer. Both the compound and its nanoformulation caused an appreciable decrease in tumor growth and hemoglobin percentages of tumor mass, signifying reduced angiogenesis with nanoformulation exhibiting more bioavailability and functional efficacy than [353]. Resveratrol combined with ginkgetin, a phytochemical obtained from Ginkgo biloba, exhibited a synergistic effect in suppressing VEGF-induced endothelial cell proliferation, migration, invasion, and tube formation in HT29 cell-induced xenografts in mice. When administered together, these two compounds demonstrated a synergistic antitumor effect with 5-FU, causing a reduction in micro vessel density of the tumors. Furthermore, the combinatorial treatment relieved the 5-FU-induced inflammatory response by lowering the expression of COX-2 and inflammatory cytokines [354]. Resveratrol also suppressed TGF-β1/Smad signaling, downregulated Snail and vimentin, and upregulated E-cadherin expression, which in turn inhibited EMT [349]. 

## 8. Phenolics in Clinical Trials for CRC Treatment

Many of the compounds discussed here, such as curcumin, resveratrol, EGCG, genistein, and fisetin, entered into different phases of clinical trials. Curcumin, the most studied phytochemical in both preclinical and clinical studies, has been tested for its effectiveness as an anti-inflammatory agent as well as its potential in prevention, management and therapy of different cancer types, including CRC [405]. The anticancer potential of resveratrol has been documented through studying its efficacy, safety, and pharmacokinetics in more than 244 clinical trials, with additional clinical trials currently being carried out by independent groups [406,407]. Although the clinical utility of resveratrol is well documented, the rapid metabolism and poor bioavailability have limited its therapeutic use [406,408]. Clinical trials on green tea extract containing EGCG as the major active component were conducted, demonstrating the good tolerance of the agent with no significant advantage of its inclusion between the placebo and the treated groups [409]. The efficacy of flavonoid fisetin supplementation on the inflammatory status and MMP levels was tested in small groups of CRC patients, while several markers were measured to assess its therapeutic efficacy, treatment with this polyphenol primarily resulted in the significant changes in IL-8 concentrations compared to the placebo group [410]. The safety and tolerability of genistein in combination with a chemotherapy agent in metastatic CRC were studied in a clinical trial with a small group of patients receiving FOLFOX or FOLFOX-bevacizumab. The results demonstrated the safety and tolerability of the treatment with notable efficacy [379]. While the results of these studies are encouraging, additional studies are needed to assess the long-term use of these phytochemicals in the clinic.

## 9. Conclusions and Future Perspectives 

CRC is the third most diagnosed and second leading cause of cancer-related death worldwide. According to recent statistics, CRC claims close to a million lives, which is about half of the population it affects globally every year. Although the CRC death rate has declined due to routine screening and early detection, CRC incidence is predicted to be doubled by the end of this decade due to various reasons, demanding an urgent need to overcome the limitations of current treatment strategies, including the development of alternative therapy regimens. This review aims to present a detailed account of the recent advances in studies on various phenolic phytochemicals with anti-CRC activities demonstrated in animal experiments with the underlying molecular basis of their actions (summarized in Table 3). 

As discussed here, the phytochemicals were reported to act through inhibiting hallmarks of various CRC attributes, such as the potential of cell growth and proliferation, self-renewal, invasion, migration, and angiogenesis through inducing apoptosis, ferroptosis, and autophagy-mediated cell death pathways (Figure 6). These activities involved the modulation of various pathways, such as the levels of proinflammatory cytokines and chemokines (IL-1, IL-6, ICAM-I, TNF, COX-2, iNOS, KC, and MCP1), oxidative stress markers and pathways (SOD, catalase, thiolase, glutathione peroxidase, GSH and Keap1/NRF2/GSK-3β/HO-1), cell cycle regulators (cyclin D1, cyclin E, and CDK 4/6), apoptotic/autophagy regulators (p21, p53, caspase-3, caspase-9, Bax, Bcl-2, Bak, and Beclin1), proliferative signaling pathways regulators (PI3K/Akt/mTOR/AMPK, Wnt/β-catenin, MAPK-p38, ERK, MEK, and c-Myc), regulators of invasion, migration, metastasis, and angiogenesis (Notch1, STAT-3, VEGF, CD31, MMP-2, MMP-3, MMP-9, MMP-16, EGFR, Twist1, Vimentin, FMS-related tyrosine kinase 4, endothelial growth receptor-3, Snail, N-cadherin, E-cadherin, TIMP-1, and TIMP-2), stemness (CD133, CD44, ALDH1, CD29, DCLK-1, and LGR5) and expression of tumor suppressive miRNAs (miR34a, miR200c, and miR145). The downregulation of COX-2 levels can be achieved upon treatment with EGCG [206], curcumin [194,197], kaempferol [239], luteolin [242,243], myricetin [259], naringenin [262], piceatannol [342], pterostilbene [344], syringic acid [326], boeravinone B [191], hesperidin [227], isoliquiritigenin [235], orientin [268], quercetin [281], and xanthohumol [301]. Caffeic acid suppressed TPA-induced activation of AP1 and NF-κB signaling [303]. Many phytophenols can induce an antioxidant response, such as EGCG, gallic acid, boeravinone B, eriodyctyol, luteolin, and morin. Caffeic acid phenethyl ester and caffeic acid phenylpropyl ester-induced mTOR inhibition through the activation of AMPK [305]. Isoangustone A upregulated AMPK phosphorylation in vivo [234]. Pterostilbene inhibited EGFR in an AOM-induced colonic adenomas in mice [344].

There is increasing evidence in favor of the idea that diet can influence the intestinal microbiome and thus the risk of CRC. Diets rich in fruits and vegetables can be associated with gut microbiome rich in Prevotella compared with Bacteroides associated with good colonic health while the consumption of diet with low plant-based food rich in processed food led to the opposite effects [411,412]. Diets rich in plant-based nutraceuticals could regulate host immune and inflammatory behavior and thus gut homeostasis through modulating the composition and functionality of the gut microbiome [413]. Therefore, CRC incidence and progression can be reduced by modulating gut microbiome by careful choice of diet and phytochemicals which could be a promising and efficient way to reduce the burden of CRC [413]. Gut microbiota can digest dietary phytochemicals by their unique ability to produce short chain fatty acids, such as butyric acid, with anti-inflammatory and antineoplastic activity [414]. Phenolic phytochemicals have served us as an important source of novel drugs/leads. While the studies discussed here provided encouraging results, several issues are needed to be considered to get a step closer to the end users, such as: 1.Apparently, the functions of many phytochemicals are limited by their poor solubility, absorption, and bioavailability. Encapsulation by nano-formulation as well as chemical derivatization of the compound could resolve this issue.2.Some cases reproducing the activity observed in preclinical animal models into the clinic/human could be challenging due to several factors. Success in this endeavor requires careful optimization in administered doses to assess functional synergy, if any, with anti-CRC regimens used in the clinic. Once positive results are obtained in the preclinical settings, testing the validity of the finding, such as safety and efficacy, in clinical trials with appropriate controls will be important to move further.3.It is reasonable to think that a phenolic compound showing very weak and toxic activity can yield desirable effect when combined with another phytochemical. Therefore, a careful combination of selected polyphenols can yield unique anti-CRC activity. It is important to clearly determine the maximum tolerable dose of a phytochemical to better understand its therapeutic efficacy alone or in combination with another phytochemical or drug.4.Once a phenolic compound with unique anti-CRC activity is identified, it would be important to develop strategies to synthesize the compound in the laboratory, given the very low abundance of a secondary metabolite in the plants. A detailed understanding of the pharmacophore responsible for the observed function should be helpful for chemical synthesis or semi-synthesis, and cellular target identification of the compound. Given the structural complexity of the plant secondary metabolites, it is often a major challenge for natural product chemists and medicinal chemists to solve. Ideally, the simultaneous engagement of experts from interdisciplinary areas, such as ethnopharmacology, molecular biology, biochemistry, natural product chemistry, medicinal chemistry, bioinformatics, and pharmacology, will be necessary to achieve progress in real-time in harvesting the full potential of natural products as the source of novel drug leads.

## Figures and Tables

**Figure 1 cancers-15-00993-f001:**
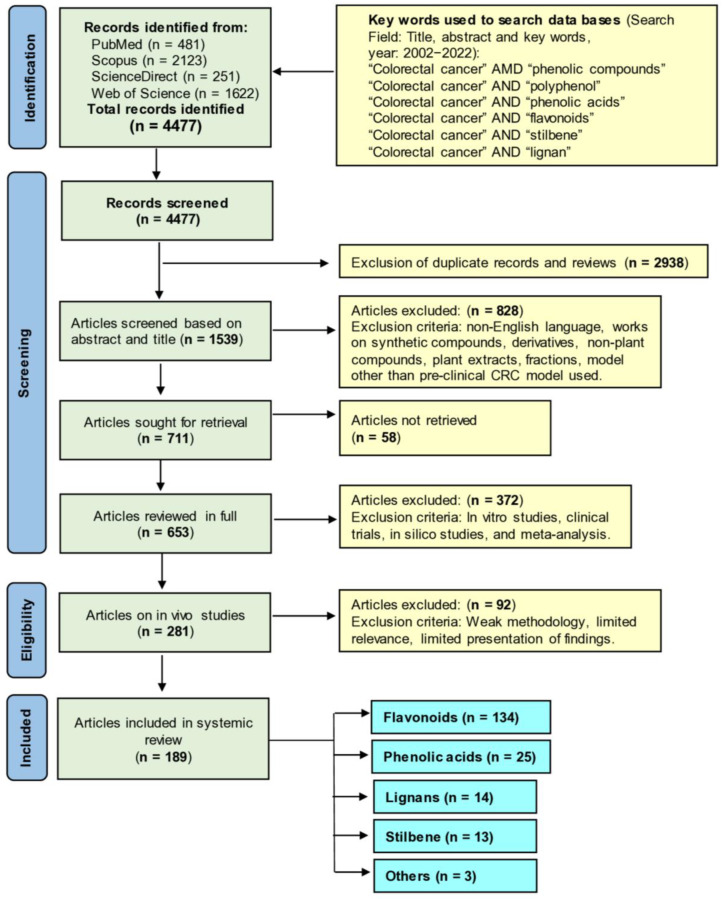
The PRISMA flow chart summarizing the literature search. Here “n” represents the number of articles.

**Figure 2 cancers-15-00993-f002:**
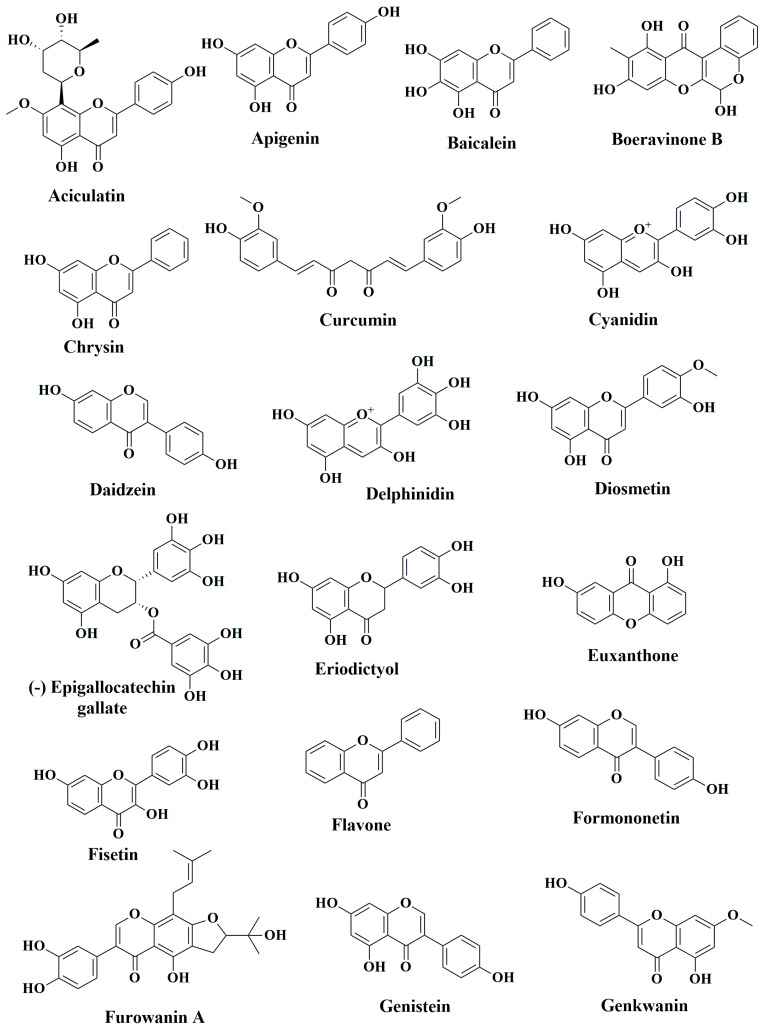
Chemical structures of flavonoids with in vivo anti-CRC activities.

**Figure 3 cancers-15-00993-f003:**
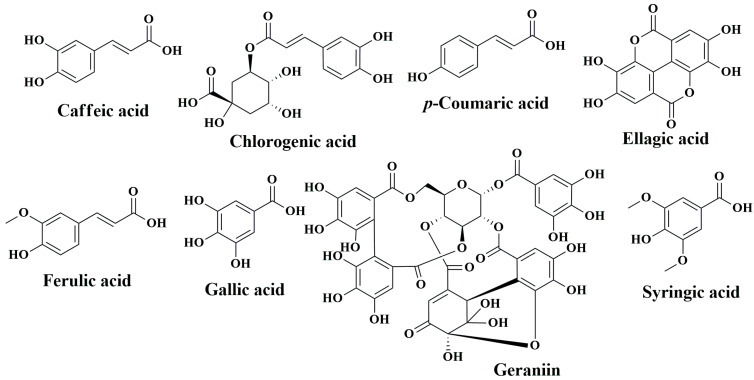
Chemical structures of phenolic acids with in vivo anti-CRC activities.

**Figure 4 cancers-15-00993-f004:**
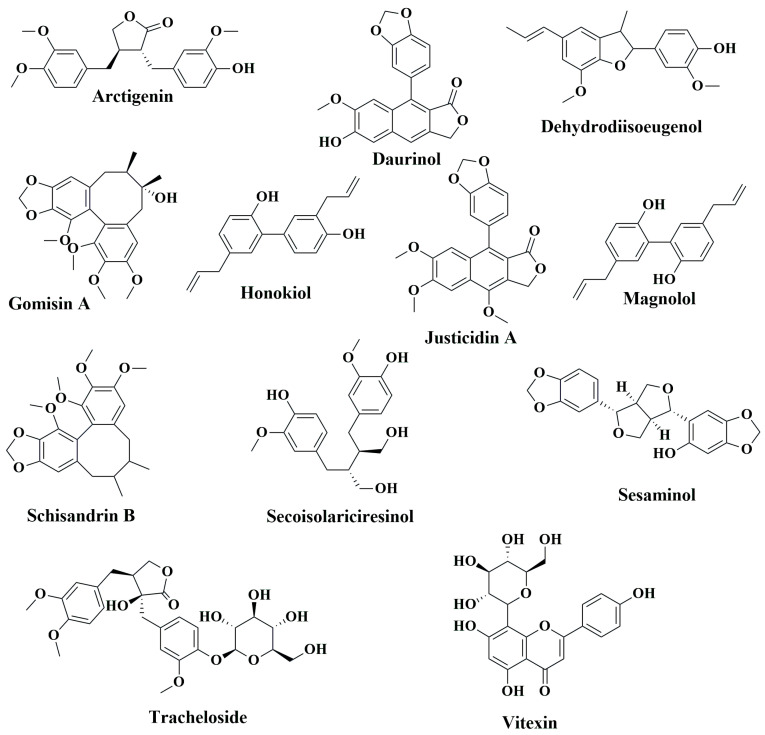
Chemical structures of lignans with in vivo anti-CRC activities.

**Figure 5 cancers-15-00993-f005:**
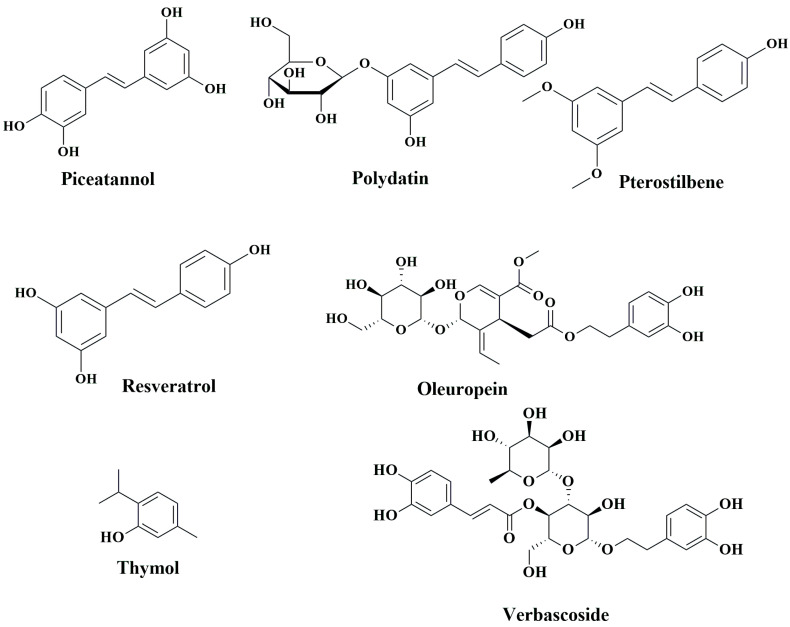
Chemical structures of stilbenes and miscellaneous compounds with in vivo anti-CRC activities.

**Figure 6 cancers-15-00993-f006:**
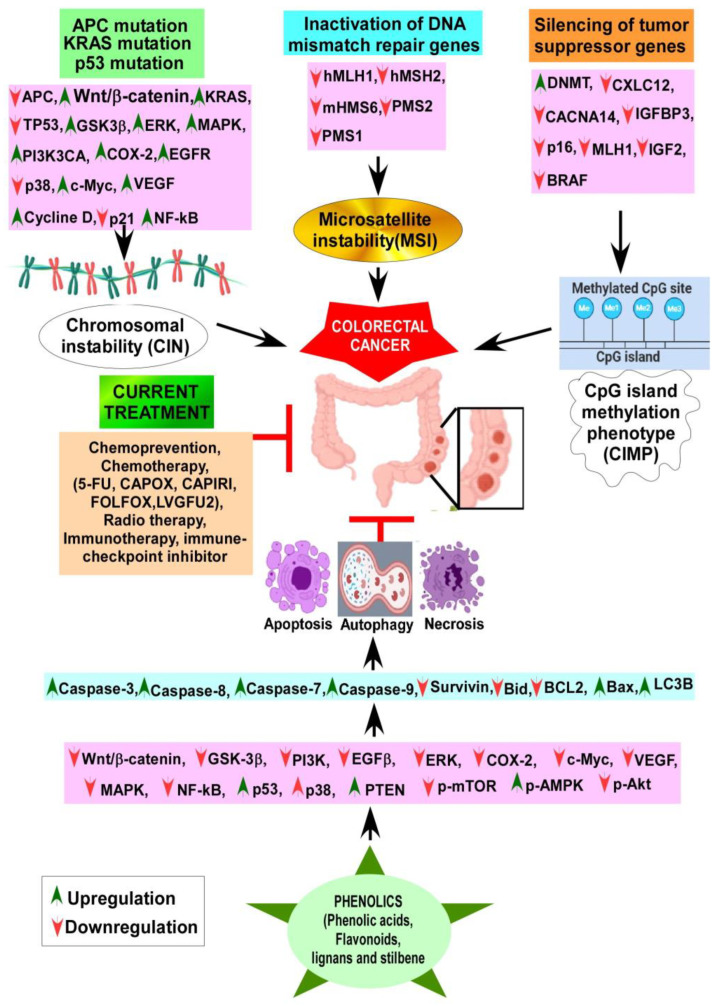
Genetic and molecular basis of colorectal cancer along with the current treatment strategies where potentials of phenolic compounds were indicated. CIN, MSI and CIMP are the prime factors in CRC development. Besides the currently available chemotherapeutic treatment strategies, different polyphenols are reported to induce CRC cell death by apoptosis and/or autophagy and/or necrosis.

**Table 1 cancers-15-00993-t001:** Genes involved in different CRC syndromes and associated clinical symptoms.

Syndrome	Genetic Defects	Clinical Manifestations	References
Hereditary nonpolyposis cancer syndromes			
Lynch syndrome	*MLH1*, *MSH2*, *MSH6*, *MSH3*, and *PMS2*	Increased risk for CRC, (10–47%) depending on gene mutated; asymptomatic unless altered bowel habits, GI bleeding due to tumors/polyps occurs; increased risk for endometrial cancer; extracolonic manifestations are associated as Muir-Torre, Turcot.	[44,46,47]
Muir-Torre syndrome (HNPCC + Sebaceous gland malignancies)	*MLH1*, *MSH2*, *MSH6,* and *PMS2*	Sebaceous skin tumor/keratoacanthoma and Lynch syndrome features.	[48,49]
Turcot syndrome type 1 (HNPCC with primary brain tumors)	*MMR*, *MLH1*, and *PMS2*	Features of Lynch syndrome + primary brain tumors.	[50,51,52,53]
Hereditary polyposis colorectal cancers			
Familial adenomatous polyposis (FAP) syndrome	*APC*	More than colorectal adenomatous polyps; 100% cancer risk	[50,54]
Turcot syndrome type II (FAP with Primary Brain tumors)	*APC*	FAP syndrome + primary brain tumors, medulloblastoma, glioblastoma, astrocytoma.	[50,51,52,53]
Gardner syndrome	*APC*	FAP syndrome+ extraintestinal manifestations of desmoid tumors; sebaceous cysts; osteomas of mandible, skull, fibromatosis, congenital hypertrophy of retinal pigment epithelium (CHRPE); adrenal adenomas.	[55,56]
Adenomatous polyposis syndromes	*APC* and *MUTYH*	Increased number of colorectal adenomas (10–100 s), serrated polyposis, mixed polyps; duodenal adenomas are common; 43–33% increased risk of CRC; increased thyroid nodules, adrenal lesions, jawbone cysts.	[50,57,58,59]
Juvenile polyposis coli	*BMPR1A* and *SMAD4*	Multiple hamartomatous polyps in the GI tract- mainly colorectum; rectal bleeding due to polyps is a common presenting symptom; anemia due to bleeding is common; extracolonic manifestations hereditary hemorrhagic Telangiectasia (HHT) telangiectasias of buccal mucosa and skin, epistaxis, and anemia, with AV malformations; colorectal cancer risk 38.7% increased.	[60,61,62]
Peutz-Jeghers syndrome	*STK11*	Mucocutaneous pigmentation; hamartomatous polyps; 39% increased risk for CRC.	[63,64]
Cowden syndrome (multiple hamartomasyndrome)	*PTEN*	Mucocutaneous lesions and macrocephaly; skin manifestations; uterine leiomyomas, ovarian cysts; multiple hamartomas on any organ; increased risk of breast, thyroid, renal, endometrial, and colorectal cancer; 9–16% risk of CRC.; increased risk for malignant melanomas; specific dysplastic gangliocytoma of the cerebellum; Lhermitte-Duclos disease is specific to Cowden disease.	[65,66]

Abbreviations: MUTYH, mutY DNA glycosylase; STK11, serine/threonine kinase; 11SMAD4, mothers against decapentaplegic homolog 4; PTEN, phosphate and tensin homolog; BMPR1A, bone morphogenic protein receptor type 1A; MLH, MutL homolog; MSH, MutS homolog; MMR, mismatch repair.

**Table 2 cancers-15-00993-t002:** Various drugs alone and in combination tested for their effects on clinical CRC chemoprevention studies.

Drugs	Study Design	Mechanism	Main Findings	References
Aspirin	Meta-analysis	COX-2 inhibition	There was a dose-dependent reduction in the risk of CR by aspirin. An aspirin dose of 75–100 mg/day reduced the risk by 10%, and 325 mg/day reduced the risk by 35% (Meta-analysis of 45 studies [RR = 0.73, 95% confidence interval (CI) 0.69–0.78])	[103,104,105,106]
Non-aspirin NSAIDS	Meta-analysis	COX-2 inhibition	Data from 23 studies suggested using higher doses of non-aspirin NSAIDs in the general population aged 40 years or older reduced CRC risk, specifically for white women, for distal colon cancer. (Pooled ODDs ratio was 0.74 (0.67–0.81), I2 = 75.9%, *p* < 0.001.)	[107]
Sulindac+ DFMO	RCT	Sulindac inhibits COX-2 DFMO- irreversibly inhibits Ornithine decarboxylase (polyamine synthesis)	Significant reduction of recurrent adenomas (12 vs. 41%, risk ratio 0.30), advanced adenomas (0.7 vs. 8.5%, risk ratio 0.09), and multiple adenomas (0.7 vs. 13.2%, risk ratio 0.06)	[108,109]
DFMO + Aspirin	RCT	Aspirin inhibits COX-2 DFMO inhibits polyamine synthesis Both combined may have a synergistic action.	After one year of treatment, in the DFMO + aspirin arm vs. placebo, there was a significant reduction in rectal aberrant crypt foci (precursor of rectal carcinoma). (74% vs. 45%, *p* = 0.020). No statistically significant reduction of colorectal adenomas was observed.	[110]
Erlotinib + Sulindac	RCT	Erlotinib is an EGFR inhibitor; sulindac is a COX-2 inhibitor.	In 82 patients of familial adenomatous polyposis, Sulindac + Erlotinib was associated with a 69.4% decrease in those with an intact colorectum compared with placebo (95% CI, 28.8−109.2%; *p* = 0.009)	[111]
Celecoxib	Meta-analysis	Selective COX-2 inhibitor, more specific for inflammation, with fewer GI side effects. Celecoxib has higher cardiovascular mortality	3 RCTs (involving 4420 patients) and 3 post-trial studies (2159) showed a significant reduction in the incidence of adenoma RR (0.67 [95% CI, 0.62–0.72] compared with placebo). There was an increased risk of cardiovascular mortality with twice dosing 400 mg celecoxib (RR 3.42 [95% CI, 1.56–7.46]). Once-a-day dosing did not show an increased CV risk. (1.01 [95% CI, 0.70–1.46]).	[112]
Clopidogrel	Case-control Study	Clopidogrel inhibits platelet aggregation via irreversible inhibition of the P2Y12 receptor	Clopidogrel decreased CRC risk in patients receiving treatment >1 year. (0.65% AOR; 95% CI, 0.55–0.78). Dual antiplatelet therapy (Clopidogrel aspirin) had the same effect as either drug is taken as monotherapy.	[113]
Metformin	Meta-analysis	Activates AMPK, inhibits mTOR pathway	Metformin users had a significantly lower incidence. CRC (RR 0.76, CI 0.69–0.84, *p* < 0.001) compared with non-metformin users. Further analysis on the overall survival of metastatic CRC patients revealed significantly higher survival rates in metformin users (HR 0.77, CI 0.68–0.87, *p* < 0.001).	[114]
UCDA	Cohort Study	Has antioxidant action. Prevents NF-κB and AP1 activity. Inhibits c-Myc	Chronic liver disease patients with UCDA have a reduced risk of colorectal cancer. UDCA use was associated with a reduced risk of CRC (hazard ratio, 0.60; 95% confidence interval [CI], 0.39–0.92).	[115]
Statin	Meta-analysis	*3-HMGCOA* reductase inhibitor decreases cholesterol synthesis. Antioxidant activity; shows pro-apoptotic effects on human CRC lines. Anti-inflammatory properties	14 studies involving 130,994 patients. In terms of post-diagnosis statin uses, the pooled HR of all-cause mortality was 0.86 (95% CI, 0.76–0.98), and the pooled HR of CSM was 0.79 (95%CI, 0.70–0.89) (Cancer-Specific Mortality).	[116,117]
Menopausal hormone therapy (combined estrogen-progestin)	Nationwide Cohort Study (Norway)	Estrogens have been proposed to alter bile acid composition, modulate colonic transit. Decrease production of mitogenic insulin-like growth factor	The current use of postmenopausal hormone therapy was associated with a decreased CRC risk. RR (for combined estrogen-progestin therapy) in oral formulations was 0.86 (95% CI 0.71 to 1.05)	[118]
Bisphosphonates	Meta-analysis	Inhibits osteoclastic bone resorption, Anti-apoptotic effect	Meta-analysis of 34 studies and 4,508,261 participants. There was a significant reduction in the risk of CRC. (RR = 0.89, 95% CI: 0.81–0.98)	[119]

Abbreviations: RCT, randomized control trial; RR, relative risk; HR, hazard ratio; OR, odds ratio, AOR, adjusted odds ratio; CI, confidence interval; DFMO, difluoromethylornithine; UCDA, ursodeoxycholic acid.

**Table 3 cancers-15-00993-t003:** Anti-CRC effects and mechanisms of action of phenolic compounds based on in vivo studies.

Phytocompound	Source	Animal Model Studied	Dose and Route of Administration	Mode of Action	Reference
Flavonoids					
2,3,5,4′-tetrahydroxystilbene-2-O-β-D-glucoside	*Polygonum multiflorum* Thunb	AOM-induced colon carcinogenesis in male F144 rats	Oral administration, 30, 150, 250 mg/kg	Decreased the number ofACF by 47–54%; suppressed tumor growth; downregulated NF-κB in nucleus and cytoplasm; downregulated CEA	[170]
4′-hydroxychalcone	Herb, teas, and spices	APC^Min/+^ mice	Oral administration, 10 mg/kg	Reduced the incidences and size of adenomas; induced apoptosis; suppressed proliferation of polyps; downregulated Ki-67; downregulated *c-Myc, Axin2* and *CD44* gene expression	[171]
Aciculatin	*Chrysopogon aciculatus*	HCT116 induced tumor xenograft SCID mice	Intraperitoneal injection, 30 mg/kg	Suppressed tumor growth without losing weight; upregulated the expression of p53 and downregulated the expression of Ki-67; induced apoptosis; arrested cells in sub G_1_ phase	[172]
Apigenin	Parsley, wheat, onions, apples, and tea plants	AOM-induced CF-1 mice and Min mice carrying mutant APC gene	Oral administration of 0.1% dietary apigenin	Reduced ACF formation and ODC activity	[173]
		Male BALB/c-nu mice	Intraperitoneal injection, 20 mg/kg	Induced apoptosis of CRC cells; upregulated FADD expression and its phosphorylation	[174]
		Male BALB/c-nu mice injected with SW480 cells	Route of administration not reported, 50 mg/kg	Elevated transgelin and downregulation of MMP-9 expression via reducing Akt phosphorylation at Ser473 and Thr308	[175]
		APC^Min/+^ mice	Oral gavage, 25 and 50 mg/kg	Reduced the number of polyps; induced of p53 activity	[176]
		Nude BALB/c mice injected with HT-29 cells	Subcutaneous injection, 35 mg/kg	Induced apoptosis; induced autophagy through inhibition mTOR/PI3K/Akt signaling pathway	[177]
		SCID mice	Oral gavage, 25 mg/kg	Suppressed prosurvival regulators Mcl-1, Akt, and ERK	[178]
		NEDD9 knock downed DLD1 cells mediated metastasis model in female athymic nude mice	Intraperitoneal injection, 20 mg/kg	Suppressed invasion, migration, and metastasis by downregulating overexpressed Neural precursor cells expressed NEDD9	[179]
Baicalein	*Scutellaria baicalensis* Georgi	AMO and DSS induced colon tumor in male ICR mice	Oral administration, 1,5 and 10 mg/kg	Restored colon length; reduced tissue inflammation.	[180]
		SW620 xenograft in BALB/c nude mice	Intraperitoneal injection, 50 mg/kg	Suppressed tumor growth by 55% without losing body weight	[181]
		CT-26 derived tumor in female BALB/c mice	Intraperitoneal injection, 20 and 40 mg/kg	Reduced tumor growth rate; downregulated TLR4 and p-IκBα protein expression; inhibited NF-κB	[182]
		HT-29 cell-induced tumor xenograft in male nude mice	Oral administration, 10 mg/kg	Suppressed tumor growth by 29.33% compared to the control group; induced apoptosis; upregulated p53 and p21	[183]
		DLD-1 tumor xenograft in BALB/c athymic nude mice	Intragastric administration, 20 mg/kg	Suppression of tumor growth; inhibition of ERK phosphorylation; downregulation of MMP-2 and MMP-9	[184]
		HCT116 tumor xenograft in NSG immunodeficient mice	Intraperitoneal injection, 50 mg/kg	Suppressed tumorigenesis; inhibited colon cancer growth; induced apoptosis and senescence	[185]
		HCT116 tumor xenograft in athymic BALB/c nude mice	Intraperitoneal injection, 80 mg/kg	Suppressed tumor growth; induced senescence; upregulated DEPP; activated Ras/Raf/MEK/ERK pathway	[186]
		HT-29 tumor xenograft in nude mice	Intraperitoneal injection, 50 and 100 mg/kg	Suppressed tumor growth	[187]
		HCT116 tumor xenograft in athymic BALB/c nude mice	Intraperitoneal injection, 100 and 200 mg/kg	Suppressed tumor growth; induced apoptosis; suppressed cancer stem cells; inhibited EMT and cyclin D1	[188]
		APC^Min/+^ mice	Oral administration, 30 mg/kg	Reduced tumor numbers; suppressed IL-1β, IL-2, IL-6, and IL-10	[189]
		HCT116 tumor xenograft in male BALB/c nude mice	Intraperitoneal injection, 100 mg/kg	Suppressed tumor growth; decreased circMYH9, mir761 and HDGF	[190]
Boeravinone B	*Boerhaavia diffusa*	DMH-induced CRC in Swiss albino Wistar rats	Intraperitoneal injection, 20 and 40 mg/kg	Decreased the number of tumor incidences; downregulated LPO; upregulated catalase, SOD and GSH; downregulated TNF-α, IL-1β, IL-6, COX-2, PGE2 and iNOS; upregulated levels of IL-4 and IL-10; down regulated MPO; downregulated the expression of GDI2 mRNA	[191]
Chrysin	*Passiflora caerulea, Passiflora incarnata, Oroxylum indicum*	AOM-induced ACF in male F344 rats	Dietary administration, 0.001% and 0.01%	Reduced mitotic index and increased apoptotic index; reduced the frequency of ACF	[192]
		Male albino rats injected with DMH + DSS	Oral administration, 125 and 250 mg/kg	Reduced the level of CXCL1, AREG and MMP-9	[193]
Curcumin	*Curcuma longa*	DSS-induced colitis in C57BL/6 mice	Oral consumption as dietary supplement, 0.6%	Reduced tumor incidences; inhibited nuclear translocation of β-catenin; downregulated TNF-α and interferon-γ; downregulated COX-2 and p53	[194]
		HCT116 tumor xenograft in female ICR SCID mice	Intragastric administration, 500 mg/kg	Suppressed tumor growth; inhibited proteasome; suppressed proliferation; induced apoptosis	[195]
		AOM-DSS induced CRC in male C57BL/6 mice	Oral gavage, 500 mg/kg	Reduced CRC tumor number; downregulated IL-1β, IL-6, COX-2 and β-catenin; suppressed Axin2 by inhibiting Wnt/β-catenin pathway	[196]
		AOM-induced colonic preneoplastic lesion in C57BL/KsJ-db/db obese mice	Dietary supplement, 0.2% and 2.0%	Inhibited colonic premalignant lesion	[197]
		HCT116 tumor xenograft in athymic nu/nu nude mice	Oral administration, 1 g/kg	Enhanced the efficacy of radiation therapy; suppressed NF-κB activity and expression	[198]
		Colo205 and LoVo tumor xenografts in athymic nu/nu mice	Tail vein injection, 40 mg/kg	Inhibited tumor growth; suppressed angiogenesis	[199]
		AOM-induced colon carcinogenesis in Il10^−/−^ mice	Oral administration, 1%	Reduced colon tumors	[200]
		AOM/DSS-induced colitis in C5757BL/6 mice	Oral administration, 25 mg/kg	Suppressed colitis-associated colon cancer and reduced tumor number	[201]
Cyanidin	Blackberries (*Rubus fruticosus*)	Apc^Min/+^ mice	Dietary supplementation, 0.03%, 0.1% or 0.3%	Reduced adenoma counts	[202]
Daidzein	Soybeans and soy-based products, and nuts	Male albino rats injected with DMH + DSS	Oral administration, 5 and 10 mg/kg	Reduced the level of CXCL1, AREG and MMP-9	[193]
Delphinidin	Berries, pomegranates, eggplant, roselle, and wine	Male BALB/c nude mice xenograft with luciferase-transfected DLD-1 cells	Intraperitoneal injection, 100 μM	Suppressed integrin/FAK nexus; elevated miR-204–3p levels	[203]
Diosmetin	Chamomile, parsley, rosemary, rooibos tea, green tea, and other plants fof the mint and citrus family (Lamiaceae)	NCr nu/nu nude mice injected with HCT-116 cells	Oral administration, 50 and 100 mg/kg	Downregulated Bcl-2; upregulated Bax	[204]
EGCG	*Camellia sinensis* L. Ktze	SW837 xenograft in male BALB/c nude mice	Oral administration, 0.01% and 0.1%	Reduced tumor growth; inhibited phosphorylation of VEGFR-2, Akt and ERK	[205]
		AOM-induced colonic premalignant lesions C57BL/KsJ-db/db mice	Oral administration, 0.01% and 0.1%	Decreased p-IGF-IR, p-GSK-3β, β-catenin, COX-2 and cyclin D1 in colonic mucosa; reduced IGF-I, insulin, triglyceride, cholesterol and leptin in serum	[206]
		AOM-induced colonic carcinogenesis in ICR mice	Oral administration, 0.25% and 0.5%	Inhibited large ACF formation; inhibited iNOS and COX-2	[207]
		HCT116-SDCSCs tumor xenograft in athymic nude mice	Cells were pretreated, 100 μM	Suppressed tumor formation; downregulated Notch1, Bmi1, Suz12, and Ezh1; upregulated miR-34a, miR-145 and miR-200c	[208]
		DMH-induced colon carcinogenesis in Wister rats	Oral administration, 0.2%	Inhibited ACF and induced apoptosis	[209]
		DMH-induced CRC in male Wistar rats	Oral administration, 50, 100 and 200 mg/kg	Lowered ACF formation; reduced tumor volume	[210]
Eriodictyol	*Eriodictyon californicum*	DMH-induced colon carcinogenesis in male albino Wistar rats	Intragastrical administration, 200 mg/kg	Suppressed the number of polyps, ACF and lipid peroxidation levels; upregulated catalase, SOD, GP_X_, GST, GSH and GR	[211]
Euxanthone	*Polygala caudata*	HT-29 cells induced tumor in BALB/c nude mice	Intraperitoneal injection, 20 and 40 mg/kg	Suppressed tumor growth; induced apoptosis; upregulated Bax; downregulated Bcl-2; induced caspase-3 cleavage; downregulated CIP2A expression and upregulated PP2A	[212]
Fisetin	Strawberry, apple, persimmon, grapes, onion, and cucumber	AOM and DSS induced CAC in male BALB/c mice	Oral administration, 20 mg/kg	Suppressed dysplastic lesions; induced apoptosis in colonic tissue; downregulated Bcl-2 and STAT3	[213]
		FC1 mice, 3K1 mice, Apc^Min/+^ males, 3K1Apc^Min/+^ mice, B6 congenic strain, B6 FC13K1Apc^Min/+^ mice	Intraperitoneal injection, 1 mg/animal	Upregulated AMPK phosphorylation; suppressed PI3K/Akt/mTOR signaling	[214]
		Male athymic nude mice	Oral administration, 400 and 800 mg/kg	Induced apoptosis, caspase-8 and cyt.; inhibited IGF1R and Akt	[215]
		CT-26 tumor in BALB/c nude mice	Subcutaneous injection, 5 mg/kg	Suppressed oncoprotein securin in p53-independent fashion	[216]
		BALB/c mice	Tail vein injection, 50 mg/kg	Inhibited programmed cell death and angiogenesis	[217]
		HCT116 tumor xenograft in mice NOD/Shi-scid-IL2R gamma (null) (NOG)	Intraperitoneal injection, 30, 60 and 120 mg/kg	Suppressed tumor growth in a dose-dependent manner	[218]
Flavone	Fruits and vegetables	DMM-induced colon carcinogenesis in C57BL/6J mice	Subcutaneous injection, 15 and 400 mg/kg	Suppressed ACF formation and multiplicity	[219]
Formononetin	*Astragalus membranaceus*	Female BALB/c-nu/nu mice injected with HCT-116 cells	Intraperitoneal injection, 20 mg/kg	Decreased VEGF, MMP-2 and MMP-9 levels	[220]
Furowanin A	*Millettia pachycarpa* Benth	HT-29 tumor xenograft in male athymic BALB/c nude mice	Intraperitoneal injection, 20 and 40 mg/kg	Suppressed tumor growth, induced apoptosis and autophagy; upregulated cleaved caspase-3, LC3BII, Beclin and p27; downregulated Ki-67, pSTAT3, Mcl-1, p62, and cyclin D	[221]
Genistein	*Genista tinctoria*	DMH-induced colon cancer in Wistar rats	Oral administration, 2.5 mg/kg	Regulated tumor microenvironment; upregulated SOD, CAT, GPx, GR, vitamin A, vitamin C, vitamin E and GSH; activated NRF2 and HO-1; reduced expression of CD133, CD44 and β-catenin	[222]
		AOM-induced colon cancer in Sprague-Dawley rats	Dietary supplementation, 140 mg/kg	Suppressed the expression of cyclin-D1 and c-Myc; decreased expression of Wnt5a, Sfrp1, Sfrp2, and Sfrp5; downregulated Wnt/ β-catenin pathway	[223]
		HCT116 tumor xenograft in athymic BALB/c mice	Oral administration, 75 mg/kg	Didn’t inhibit tumor growth; suppress metastasis; downregulated MMP-2 and EGFR3	[224]
Genkwanin	Dried flower buds of *Daphne genkwa*	APC^Min/+^ mice	Oral administration, 12.5 and 25 mg/kg	Inducted host defense; reduced proinflammatory cytokine levels	[225]
		AOM/DSS-induced C57BL/6J mice	Oral administration, 22.5 mg/kg	Suppressed colon cancer growth by triggering tumor cell death; inhibited of pro-inflammatory cytokines	[226]
Hesperidin	Citrus fruits	AOM-induced Swiss albino mice	Oral administration, 25 mg/kg	Inhibited NF-κB, iNOS and COX-2; reduced cellular oxidative indicators and improved antioxidant status	[227]
		AOM-induced male Swiss albino mice	Oral administration, 25 mg/kg	Inhibited the constitutively active Aurora-A driven PI3K/Akt/GSK-3 and mTOR; activated autophagy	[228]
		AOM-induced male F344 rats	Oral administration, 1000 ppm	Inhibited ACF formation; reduced colonic mucosal ODC activity and polyamine levels in the blood	[229]
		DMH-induced CRC in albino rats	Oral administration, 25 mg/kg	Elevated the expression of Smad4 and activin A	[230]
Hinokiflavone	*Selaginella tamariscina*, *Juniperus phoenicea*, and *Rhus succedanea*	CT-26 tumor in female BALB/c mice	Intraperitoneal injection, 25 and 50 mg/kg	Suppressed tumor growth and proliferation; induced apoptosis; downregulated Ki-67 and MMP-9	[231]
Icariside II	*Epimedi* Herba	SW620 tumor xenograft in nude BALB/c mice	Intraperitoneal injection, 25 mg/kg	Suppressed tumor growth; induced apoptosis	[232]
Icaritin	*Epimedium* sp.	HT-29 tumor xenograft in male nude mice	Oral gavage, 10 mg/kg	Suppressed tumor growth and volume	[233]
Isoangustone A	*Glycyrrhiza* sp.	SW480 tumor xenograft in male BALB/c nu/nu mice	Intraperitoneal injection, 10 mg/kg	Suppressed tumor growth; induced autophagic cell death; upregulated phosphorylation of AMPK, ACC and LC3B-1 and II levels	[234]
Isoliquiritigenin	*Glycyrrhiza glabra*	AOM/DSS-induced colon carcinogenesis in male BALB/c mice	Intragastrical administration, 3, 15 and 75 mg/kg	Suppressed tumorigenesis; inhibited macrophage polarization; upregulated TNF-α, INF-γ and IL-12; downregulated TGF-β, IL-10 and IL-1 and COX-2	[235]
	*Glycyrrhiza uralensis* Fisher	AOM-treated colon carcinogenesis in 344 rats	Oral administration, 100 ppm dietary supplementation	Suppressed ACF formation; induced apoptosis	[236]
Isorhamnetin	*Opuntia ficus-indica*	HT-29 RFP xenograft in immunosuppressed mice	Oral administration, dose not reported	Elevated cleaved caspase-9, Hdac11, and Bai1 proteins	[237]
		FVB/N mice treated with AOM/DSS	Oral administration, dietary supplement, dose not reported	Inhibited nuclear translocation of β-catenin and c-Src stimulation; activated CSK	[238]
Kaempferol	Apple, tea, broccoli, and grapefruit	DMH-induced colorectal carcinogenesis in male Wistar rats	Oral administration, 200 mg/kg	Restored CAT, SOD, and GPx	[239]
		DMH-induced colon carcinoma in male Sprague Dawley rats	Oral administration, 200 mg/kg	Reduced multiple plaque lesions and preneoplastic lesions	[240]
		DMH-induced colitis in Sprague-Dawley albino rats	Oral administration, 200 mg/kg	Reduced multiplicity of the ACF; downregulated COX-2 and PCNA	[241]
Luteolin	Celery, parsley, broccoli, onion leaves, carrots, peppers, cabbages, and tea	DMH-induced carcinogenesis in male Wistar rats	Subcutaneous injection, 0.2 mg/kg	Reduced the number of tumor polyps and colon polyploids; decreased COX-2 level in blood and colonic tissue	[242]
		AOM-induced CRC in male BALB/c mice	Oral administration, 1.2 mg/kg	Reduced the levels of alkaline phosphatase and lactate dehydrogenase; suppressed iNOS and COX-2	[243]
		AOM-induced CRC in male BALB/c mice	Oral administration, 1.2 mg/kg	Reduced cytochrome b_5_, cytochrome P450 and cytochrome b_5_; enhanced the expression of UDP-GT and GST in colonic tissue; upregulated Nrf2	[244]
		CT-26 mediated lung metastasis	Oral administration, 10 and 50 mg/kg	Suppressed lung nodules and nodule volume; inhibited MMP-9 expression	[245]
		AOM-induced colon carcinogenesis in BALB/c mice	Oral administration, 1.2 mg/kg	Inhibited MMP-2 and MMP-9; downregulated γ-glutamyl transferase, 5′ nucleotidase, cathepsin D, and carcinoembryonic antigen	[246]
		HT-29 tumor xenograft in BALB/c nude mice	Intragastric administration, 100 mg/kg	Suppressed CRC metastasis; upregulated miR-384; downregulated pleiotrophin expression	[247]
		HT-29 tumor xenograft in BALB/c nude mice	Intraperitoneal injection, 50 mg/kg	Inhibited tumor growth; induced apoptosis	[248]
Lysionotin	*Lysionotus pauciflorus* Maxim	HCT116 tumor xenograft in athymic nude mice	Intraperitoneal injection, 20 mg/kg	Suppressed tumor growth; induced ferroptosis	[249]
Magnolin	*Magnolia biondii*	HCT116 tumor xenograft in female BALB/c athymic nude mice	Intraperitoneal injection, 20 mg/kg	Suppressed tumor growth; downregulated LIF, STAT3 and Mcl-1	[250]
Morin	Old fustic (*Chlorophora tinctoria*) and osage orange (*Maclura* *pomifera*)	Male athymic nude mice injected with HCT-116 cells	Intraperitoneal injection, 30 and 60 mg/kg	Inactivated NF-κB signaling	[251]
		Male albino Wistar rats injected with DMH	Intraperitoneal injection, 30 and 60 mg/kg	Modulated tumor metabolism via β-cateinin/c-myc signaling, glycolysis and glutaminolysis pathways	[252]
		Pirc rats (F344/NTac-Apc am1137)	Dietary supplementation, 50 mg/kg	Restored the sensitivity to apoptosis by inhibiting LMW-PTP	[253]
		Male albino Wistar rats injected with DMH	Intragastric administration, 50 mg/kg	Reduced ACF formation; suppressed fecal and mucosal biotransformation enzymes	[254]
		Male albino Wistar rats injected with DMH	Intragastric administration, 50 mg/kg	Inhibited NF-κB and inflammatory mediators; suppressed proapoptotic pathway	[255]
		DMH-induced colon carcinogenesis in a male Wistar rats	Oral administration, 50 mg/kg	Reduced lipid hydroperoxides and CD; increased superoxide SOD, CAT, GST, GPx, GR; decreased GSH	[256]
Myricetin	Tea, barriers, fruits, vegetables	DMH-induced rat colon carcinogenesis	Dietary supplementation, 50, 100 and 200 mg/kg myricetin	Restored CAT, GPx and GSH	[257]
		APC^Min/+^ C57BL/6J mice	Oral gavage, 100 mg/kg	Promoted apoptosis in adenomatous polyps; lowered IL-6 and PGE2; downregulated p38 MAPK/Akt/mTOR signaling pathway	[258]
		AOM/DSS-induced in BALB/c mice	Oral gavage, 40 and 100 mg/kg	Inhibited the development of colorectal tumors and colorectal polyps; decreased the levels of TNF-, IL-1, IL-6, NF-κB, p-NF-κB, COX-2, PCNA, and cyclin D1	[259]
		AOM/DSS-induced colitis in C57BL/6 mice	Oral administration, 100 mg/kg	Decreased CSF/M-CSF, IL-6, and TNF-α in colonic mucosa; inhibited NF-κB/IL-6/STAT3 pathway	[260]
Naringenin	Oranges, lemons, and grapefruit	AOM-induced colon carcinogenesis in rats	Dietary supplement, 0.02%	Reduce the number of HMACF by 51% and the proliferative index by 32%	[261]
		DSS-induced murine colitis model	Oral administration, 50 mg/kg	Decreased iNOS, ICAM-1, MCP-1, COX-2, TNF-α, and IL-6 transcript levels	[262]
		HT-29 tumor xenograft in athymic NIH Swiss nude mice	Oral administration, 40 mg/kg	Suppressed tumor growth; inhibited COX-1	[263]
Naringin	Oranges, lemons, and grapefruit	DMH-induced female Wistar rats	Oral gavage, 10, 100, 200 mg/kg	Reduced cell proliferation and tissue iron levels; upregulated antioxidant mineral levels	[264]
		AOM/DSS-induce Male C57BL/6 mice	Oral gavage, 50 and 100 mg/kg	Suppressed ER stress-induced autophagy in colorectal mucosal cells	[265]
		AOM-induced ACF in Sprague Dawley rats	Oral administration, 200 mg/kg	Reduced total number of ACF; suppressed proliferation; induced apoptosis; downregulated COX-2 and iNOS	[266]
Nobiletin	Peel of various *Citrus* fruits	AOM-DSS-induced colon carcinogenesis in male CD-1 mice	Oral, dietary supplement, 100 ppm	Reduced tumor incidences and multiplicity	[267]
Orientin	*Ocimum sanctum*	DMH-induced CRC in male Wister rats	Intraperitoneal injection, 10 mg/kg	Reduced NF-κB, TNF-α and IL-6; downregulated Ki-67 and PCNA; suppressed iNOS and COX-2	[268]
		DMH-induced CRC in male Wister rats	Intraperitoneal injection, 10 mg/kg	Suppressed ACF and crypt multiplicity; elevated the level of antioxidants; downregulated phase I enzymes and upregulated phase II enzymes	[269]
Oroxylin A	*Scutellaria baicalensis*	AOM-DSS induced CRC in C57BL/6 mice	Dietary supplementation, 50, 100 and 200 mg/kg	Suppressed tumor formation and colitis associated CRC; induced apoptosis; downregulated IL-6, IL-1β, p-STAT3, cyclin D, and Bcl-2; upregulated Bax	[270]
		HCT116 tumor xenograft in male athymic BALB/c nude mice and AOM-DSS induced colon carcinogenesis in male C57BL/6 mice	Oral administration, 150 and 300 mg/kg	Suppressed carcinogenesis and primary colon cancer progression; reduced triglyceride; downregulated HIF1α, Srebp1, FASN, ADRP and FABP7; upregulated CPT1	[271]
Pectolinarigenin	*Cirsium chanroenicum*	Murine CT26 CRC cells were introduced into BALB/C mice	Intraperitoneal injection, 25 and 50 mg/kg	Induced apoptotic death of cancer cells; suppression STAT3	[272]
Peonidin	Sweet potato (*Ipomoea batatas*)	AOM-induced CF-1 mice	Dietary supplementation, 10 to 30%	Blocked cell cycle at the G1 phase; activated caspase-3	[273]
Petunidin	*Lycium ruthenicum*	Nude mice	Intraperitoneal injection, 25 and 50 mg/kg	Induced ferroptosis via inhibiting SLC7A11	[274]
Phloretin	*Manchurian apricot*	COLO 205 cells derived tumor in BALB/c nude mice	Route of administration not reported, 25 mg/kg	Inhibited tumor growth; upregulated p53, p21 and E-cadherin	[275]
Polyphenon E		AOM-induced colon carcinogenesis in F344 rats	Oral administration, 0.24%	Induced apoptosis; decreased eicosanoid, prostaglandin E2, and interleukin B4 in plasma; decreased nuclear β-catenin and increased expression of RXRα,β and γ in adenocarcinomas	[276]
Procyanidin	Cider apple (*Malus domestica*)	AOM-induced Wistar rats	Oral administration, 0.01%	Suppressed protein kinase; down-regulated of polyamine production; stimulated caspase-3	[277]
		Male C57/BL6 mice transfected with CT26 cells	Oral gavage, 30 mg/kg	Reduced cellular oxidative stress through modulation of Nrf2/ARE signaling	[278]
Quercetin	Apples, nuts, cauliflower, cabbage, onions, grapes, berries, broccoli, citrus fruits, cherries, green tea, and coffee	AOM-induced colon tumor in C57BL/6J male mice	Dietary supplementation, 0.5%	Induced apoptosis; upregulated CB1-R; downregulated STAT3 and p-STAT3; downregulated Bax/Bcl-2 ratio	[279]
		Subcutaneous DLD-1 human colon tumor fragment implant in male athymic nu/nu mice	Intraperitoneal injection, 30 mg/kg	Enhanced radiosensitivity by inhibiting ATM-mediated signaling pathway	[280]
		AOM-induced CRC in male weanling Sprague-Dawley rats	Dietary supplement, 4.5 g/kg	Reduced the number of crypts; inhibited proliferation; induced apoptosis; suppressed COX-1, COX-2 and iNOS	[281]
		AOM/DSS induced colon carcinogenesis in C57BL/6J mice	Dietary supplementation, 30 mg/kg	Reduced number and size of colon tumors; suppressed inflammation; downregulated LOP, NO, SOD, G6PD, and GSH	[282]
		CT-26 lung tumor metastasis in BALB/c mice	Intraperitoneal injection, 50 mg/kg	Suppressed lung metastasis; induced apoptosis	[283]
		HT-29 tumor xenograft in BALB/c nude mice	Subcutaneous injection, 10 mg/kg	Enhanced radiosensitivity; inhibited Notch-1 signaling	[284]
Rutin	Buckwheat, Mez, Labisia pumila, *Sophora japonica* L., Schum, *Canna indica* L., and *Ruta graveolens* L.	SW480 cell-induced tumor xenograft	Intraperitoneal injection, 20 mg/kg	Suppressed tumor growth; decreased angiogenesis and VEGF levels	[285]
Scutellarin	*Scutellaria barbata*	AOM/DSS-induced male C57BL/6 mice	Intraperitoneal injection, 25, 50 and 100 mg/kg	Inhibited Wnt/β-catenin signal transduction	[286]
		RKO cells were subcutaneously implanted into female nude mice	Intraperitoneal injection, 50, 150 and 300 mg/kg	Suppressed tumor growth and metastasis	[287]
		AOM/DSS-induced mice	Intraperitoneal injection, 25, 50 and 100 mg/kg	Suppressed the Hedgehog signaling cascade	[288]
Silibinin	*Silybum marianum*	LoVo cell deposition on eight days old fertilized chicken egg	Route of administration not reported, 9.64 μg/mL	Decreased in VDI; upregulated *Flt-1* gene	[289]
		AOM-induced CRC in male Wistar rats	Intragastric intonation, 300 mg/kg	Suppressed preneoplastic lesion formation; activated apoptosis; registered sub G0/G1 cell cycle arrest; reduced MMP-7, IL-1β and TNF-α	[290]
Tangeretin	Peel of citrus fruits	HT-29 induced tumor xenograft in BALB/c nude mice	Route of administration not reported, 5 mg/kg	Suppressed tumor growth	[291]
Taxifolin	Olive oil, grapes, citrus fruits, and onion	HCT116 tumor xenograft in athymic male nude mice	Intraperitoneal injection, 15 and 25 mg/kg	Suppressed tumor growth; induced apoptosis; inhibited cyclin D; degraded β-catenin; inhibited of Akt phosphorylation	[292]
Tricin	Rice bran, oats, barley, and wheat	Colon26-Luc colon tumor and lung metastasis model in BALB/c mice	Oral gavage, 19 and 37.5 mg/kg	Suppressed tumor growth; reduced metastasis incidence	[293]
		AOM-DSS induced CRC in male Crj: CD-1 mice	Dietary supplement, 50 and 250 ppm	Restored colonic length; reduced number of incidences and multiplicity of adenomas and adenocarcinomas; downregulated PCNA and TNF-α	[294]
Troxerutin	Tea and coffee	DMH-induced colon carcinogenesis in male albino Wistar rats	Oral administration, 12.5, 25 and 50 mg/kg	Lowered ACF formation and crypt multiplicity; reduced cytochrome P450, cytochrome b_5_, cytochrome P4502E1, NADPH-cytochrome P450 reductase, and NADH-cytochrome b_5_ reductase and upregulates phase GST, DTD and UDPGT	[295]
Vitexin	Passionflower, bamboo leaves, pearl, and millet	HCT116 tumor xenograft in nude BALB/c mice	Oral administration, 25, 50 and 100 mg/kg	Suppressed tumor growth; increased phosphorylation of JNK; upregulated LC3 II and ApoL1	[296]
		HCT116^DR^ tumor xenograft in female athymic BALB/c nude mice	Oral administration, 25 and 50 mg/kg	Suppressed tumor growth; induced apoptosis; downregulated HSP90, HSP70, HSP27, Atg7, Beclin-1, LC3 II and Bcl-2; upregulated Bax and PARP1; cleaved caspase-3 and caspase-9	[297]
Wogonin	*Scutellaria baicalensis, Scutellaria radix*	AOM/DSS-induced colitis related colon cancer in C57BL/6 mice	Gastric intubation, 60 mg/kg	Decreased cell proliferation; lowered the expression and secretion of IL-6 and IL-1β and expression of NF-κB; increased Nrf2 nuclear translocation	[298]
		AOM-DSS-induced CRC animal model in C57BL/6 mice	Route of administration not reported, 50 and 100 mg/kg	Reduced tumor multiplicity; reverted colon length to normal	[299]
		SW480 induced tumor xenograft in BALB/c nude mice	Intraperitoneal injection, 2 mM	Downregulated of YAP-1 and IRF3; upregulated p-YAP	[300]
Xanthohumol	*Humulus lupulus*	AOM-induced colorectal carcinogenesis in male Sprague-Dawley rats	Oral gavage, 5 mg/kg	Suppressed tumor growth; induced apoptosis; suppressed COX-2 and iNOS	[301]
Zapotin	Tropical fruit zapote blanco (*Casimiroa edulis*)	AOM/DSS-induced female CF-1 mice	Intragastric administration, 5 and 10 mg/kg	Induced cell cycle arrest and apoptosis	[302]
Phenolic acids					
Caffeic acid	Coffee, wine tea	CT-26 lung metastasis in BALB/c mice	Oral administration, 0.1 and 0.5 g/kg	Inhibited lung metastasis; suppressed MEK1, TOPK, and TAP-induced activation of AP1, NF-κB and ERK signaling; inhibited TAP, EGF and H-Ras induced neoplastic transformation	[303]
		HCT116 tumor xenograft in NSG mice	Intraperitoneal injection, 10 mg/kg	Inhibited CSC growth and self-renewal by inhibition of PI3K/Akt signaling	[304]
		HCT116 tumor xenograft in BALB/c AnN Foxn-1 nude mice	Oral administration, 50 nmol/kg	Inhibited PI3K/Akt/mTOR pathway; suppressed MMP-9, cyclin D1, Cdk4, cyclin E, PCNA, FASN c-Myc, and N-cadherin expression; upregulated p21	[305]
		HT-29 tumor xenograft in BALB/c nude mice	Intragastric administration, CAPE (10 mg/kg); CAPE-pNO2 5, (10 and 20 mg/kg)	Inhibited tumor growth and VEGF expression; upregulated p53, p27, p21, cyt. c, and cleaved caspase-3; downregulated procaspase-3, Cdk2, and c-Myc;	[306]
		HCT116 tumor xenograft in nude mice	Oral administration, 0.2 and 2 mg/kg	Suppressed tumor growth; displayed cell cycle arrest in S phase and autophagic cell death	[307]
Chlorogenic acid	Apple, betel, coffee beans, kiwi, grapes, eggplant, pear, plum, potato, and tea	MAM acetate-induced carcinogenesis hamsters	Oral administration, 0.025% dietary supplement	Reduced colon tumor incidences; registered antioxidative effect; inhibited the activity of microsomal enzyme	[308]
		AOM-induced ACF in colon of male F344 rats	Oral administration, 0.025% dietary supplement	Reduced ACF formation and growth	[309]
Ellagic acid		AOM-induced colon tumors in rats	Oral administration, 250, 2500 and 5000 ppm	Inhibited the incidence of adenocarcinomas in the small intestine	[310]
		DMH-induced colon cancer in rats	Oral administration, 60 mg/kg	Lowered the frequency of ACF and lipid peroxidation; increased the activity of CAT, SOD, GPx, GR and GST; restored the levels of vitamin C, vitamin E and GSH	[311]
		DMH-induced colon cancer in Wistar albino rats	Oral administration, 60 mg/kg	Inhibited NF-κB, iNOS, COX-2, TNF-α and IL-6; restored the levels 5′-ND, γ-GT, CEA, AFP and LDH	[312]
		DMH-induced colon cancer in rats	Oral administration, 60 mg/kg	Inhibited PI3K-p58 activation; downregulated Akt and Bcl-2; upregulated Bax	[313]
		DMH-induced colorectal cancer in rats	Oral administration, 60 mg/kg	Inhibited ACF formation; increased the activity of CAT, SOD, GPx, and GR; inhibited ODC expression through inhibition of c-MYC	[314]
		DMH-induced colon cancer in male Laca mice	Oral administration, 10 mg/kg	Restored colon membrane alterations	[315]
Ferulic acid	Rice, wheat, pineapple, grains, and peanuts	AOM-induced colon cancer in male Fischer 344 rats	Oral administration, 250 ppm and 500 ppm	Reduced number and size of adenomas; increased the activity of GST and QR	[316]
		AOM-induced colon carcinogenesis in F344 rats	Dietary supplement of 3-(4′-geranyloxy-3-methoxyphenyl)-2propenoate (geranylated derivative of ferulic acid) 0.1% and 0.2%	Decreased the number of ACF	[317]
Gallic acid	Barriers and pomegranates	DMH-induced colon cancer in male Wister rats	Oral administration, 50 mg/kg	Reduced lipid peroxidation, LOOH, CD, SOD, CAT, GSH, GR and GPx; reduced ascorbic acid and tocopherol levels	[318]
		SW480 induced tumor xenograft in NOD SCID gamma NSG mice	Intraperitoneal injection, 200 mg/kg	Exerted antitumor activity mediated by interaction with G-quadruplexes	[319]
		DSS-induced acute colitis in C57BL/6 mice	Oral administration, 5 and 25 mg/kg	Suppressed acute colitis; inhibited phosphorylation of STAT3	[320]
		HCT116 and HT-29 tumor xenografts in BALB/c nude mice	Intraperitoneal injection, 80 mg/kg	Suppressed p-SRC, p-EGFR, p-Akt and p-STAT3	[321]
		Ulcerative colitis in rats	Oral administration, 10 mg/kg	Suppressed colon cancer; induced ferroptosis	[322]
		DMH-induced colon cancer in male albino Wister rats	Oral administration, 50 mg/kg	Elevated the activity of cytochrome P450, cytochrome b5, GST, DT-diaphorase and γ-GT	[323]
Geraniin	*Phyllanthus amarus*	SW480 tumor xenograft in nude mice	Oral administration, 10, 20 and 40 mg/kg	Suppressed tumor growth; induced apoptosis; inhibited phosphorylation of PI3K and Akt	[324]
p-Coumaric acid	Mushrooms, apples, pears, barriers, oranges, and beans	DMH-induced colon carcinogenesis in male albino Wistar rats	Intragastric intubation, 100 mg/kg	Reduced ACF, DACF, MDF and BCAC	[325]
Syringic acid	Olives, dates, pumpkins, grapes, and palms	DSS-induced mice	Oral administration, 25 mg/kg	Decreased the level of iNOS, COX-2, TNF-α, IL-1β and IL-6; reduced activation and accumulation of p-STAT-3 by decreasing expression of p65-NF-κB	[326]
		DMH-induced colorectal cancer in male rats	Oral administration, 50 mg/kg	Reduced tumor incidences, tumor volume and average number of tumors	[327]
Lignans					
Arctigenin	*Arctium lappa, Forsythia suspensa.*	CT-26 cells derived lung metastasis model in BALB/c mice	Oral gavage, 50 mg/kg	Reduced the number of lung nodules; induced apoptosis in lung tissue; inhibited EMT in lung tissue; induced cleavage of caspase-3, caspase-9, and PARP; downregulated Bcl-2 and Bcl-xL; upregulated Bax	[328]
Daurinol	*Haplophyllum dauricum*	HCT116 tumor xenograft in athymic BALB/c (*Slc*/*nu*) nude mice	Oral administration, 5 and 10 mg/kg	Suppressed tumor growth; upregulated p-Chk1(Ser345)/Chk1	[329]
Dehydrodiisoeugenol	*Myristica fragrans* Houtt	HCT116, zsw480, and patient-derived xenograft in female NOD/SCID mice	Intraperitoneal injection, 40 mg/kg	Suppressed tumor growth by inducing ER stress; upregulated BiP, PERK, and IRE1α	[330]
Gomisin A	*Schisandra chinesis*	CT-26 induced lung metastasis in female BALB/c mice	Intraperitoneal injection, 50 mg/kg	Suppressed lung metastasis; reduced the number of lung nodules; increased phosphorylation of AMPK and p38 in lung tissue	[331]
Honokiol	*Magnolia grandiflora*	SW620 tumor xenograft in female athymic BALB/c nude mice nu/nu	Intragastric administration, 50 mg/kg	Inhibited proliferation of CRC; upregulated TGF-β1 and p53 by upregulating BMP7	[332]
Justicidin A	*Justicia procumbens*	HT-29 tumor xenograft in NOD-SCID mice	Oral administration, 6.2 mg/kg	Suppressed tumor growth; induced autophagy in tumor tissue; induced apoptosis	[333]
Magnolol	*Magnolia officinalis*	CT-26 and HT-29 tumor in BALB/c and Cg-Foxn1^nu^/CrlNarl nude mice respectively	Route of administration not reported, 50 and 100 mg/kg	Inhibited tumor growth; induced apoptosis; upregulation of Fas, Fas-L, cleaved caspase-3, cleaved caspase-9 and cleaved caspase-8; downregulated NF-κB, PKCδ, ERK, Akt, C-FLIP, and MCL-1; inhibition of PKCδ-NF-κB signaling	[334]
		HCT116 tumor xenograft in female BLB/c nude mice	Intraperitoneal injection, 5 mg/kg	Suppressed tumor growth without showing any toxicity	[335]
Schisandrin B	*Schisandra chinensis, Schisandra propinqua, and Schisandra rubriflora*	AOM-DSS-induced CRC in C57BL/6 mice	Oral administration, 3.7–30 mg/kg	Suppressed SIRT1	[336]
Secoisolariciresinol	*Fitzroya cupressoides* and *Crossosoma bigelovii*	HCT116 tumor xenograft in male BALB/c nude mice	Route of administration not reported, 50, 100 and 200 mg/kg	Inhibited tumor growth; induced pyroptosis; downregulated Ki-67; upregulated N-GSDMD	[337]
		DSS-induced colitis in mice	Dietary supplementation, 200 mg/kg	Suppressed tumor growth; reduced IL-1β, IL-18, TNF-α and NLRP1	[338]
Sesaminol	*Sesamum indicum*	Ethanol-induced CRC in male C57BL/6NCr mice	Oral administration, 2.5 mg/mice	Reduced colonic lesions; downregulated iNOS and CYP2E1; lowered TNF-α, IL-6, MCP-1 and NF-κB levels; increased cell adhesion by upregulation of ZO-1, occludin and cladulin-1	[339]
Tracheloside	*Carthamus tinctorious* L. (safflower)	CT-26 lung metastasis in BALB/c mice	Oral administration, 25 and 50 mg/kg	Suppressed lung metastasis; induced apoptosis; upregulated E-cadherin RNA; downregulated N-cadherin, vimentin, snail and twist RNA	[340]
Vitexin	*Vitex negudo*	HCT116 tumor xenograft in female nu/nu mice	Intraperitoneal injection, 40 mg/kg	Inhibited tumor growth and lowered tumor volume; upregulated PUMA and p53; induced PUMA-mediated apoptosis	[341]
Stilbenes					
Piceatannol	Red and white grapes	AOM/DSS-induced colon tumor in C57BL/6J mice	Oral administration, 5 and 12.5 mg/kg	Decreased tumor number and size; decreased Ki-67- and COX-2-positive cell number; downregulated MCP1 and PD1	[342]
Polydatin	*Picea sitchensis*	Caco-2 tumor xenograft in C57BL/6 mice	Subcutaneously into the tumor, 150 mg/kg	Suppressed tumor growth; upregulated miR-382; downregulated PD-L1	[343]
Pterostilbene	Blueberries and cranberries	AOM-induced colonic ACF preneoplastic lesions and adenomas in male ICR mice	Oral administration, 50 or 250 ppm	Reduced ACF and adenoma formation; induced apoptosis; downregulated iNOS and COX-2; inhibited Wnt/β-catenin signaling through suppressing phosphorylation of GSK3β; inhibited VEGF, cyclin D, MMPs, Ras, PI3K/Akt and EGFR	[344]
		AOM-induced colon tumors in F344 rats	Oral administration, 40 ppm	Reduced the proliferation of non-metastatic adenomas; downregulated IL-1β, IL-4, TNF-α, PCNA, β-catenin and cyclin D and p-NF-κB/p65	[345]
		AOM-induced colon tumor in male BALB/c mice	Oral administration, 50 or 250 ppm	Reduced NF-κB through inhibition of PKC-β phosphorylation; downregulated iNOS, COX-2 and aldose reductase; upregulated HO-1, GR and Nrf2 signaling	[346]
		CL187 transplantation model in BALB/c nude mice	Intraperitoneal injection, 25, 50, 100 and 200 mg/kg	Inhibited Top1-mediated DNA damage repair pathway	[347]
		AOM-induced colonic ACF preneoplastic lesions in F344 rats	Oral administration, 40 ppm	Inhibited ACF formation; blocked cell proliferation and iNOS	[348]
Resveratrol	Grapes, blueberries, raspberries, mulberries, and peanuts	LoVo cell-mediated metastasis model in mice	Intragastric administration, 50, 100 and 150 mg/kg	Inhibited metastasis; decreased tumor size; suppressed TGF-β1/Smad pathway; downregulated Snail and vimentin; upregulated E-cadherin expression	[349]
		APC^CKO^/Kras^mut^ mice	Dietary supplementation, 150 ppm and 300 ppm	Suppressed tumor formation; reduced tumor size; downregulated Kras expression	[350]
		DSS-induced colon carcinogenesis in rats	Oral administration, 60 mg/kg	Reduced ACF and MDF	[351]
		HCT116 tumor xenograft in ICR SCID mice	Oral administration, 150 mg/kg	Suppressed tumor growth; induced apoptosis; inhibited NF-κB	[352]
		COLO250-luc tumor xenograft in athymic mice	Injection in tumor, 6 μg/implant	Suppressed tumor growth	[353]
		HT-29 tumor xenograft in BALB/c nu/nu mice	Intragastric administration, 480, 960 and 1920 mg/kg	Suppressed VEGF-mediated angiogenesis	[354]
Miscellaneous compounds					
Oleuropein	Olives (*Olea europaea*)	AOM-induced CRC in female A/J mice	Dietary supplementation, 125 mg/kg	Suppressed preneoplastic lesions; lowered tumor incidences; prevented DNA damage	[355]
Thymol	*Thymus vulgaris* L.	HCT116 tumor xenograft and lung metastasis model in BALB/c nude mice	Intraperitoneal injection, 75 and 150 mg/kg	Induced apoptosis; upregulated E-cadherin; downregulated N-cadherin; suppressed lung metastasis by inhibiting Wnt/β-catenin pathway	[356]
Verbascoside	Genus, *Cistanche*	HCT116 tumor xenograft in BALB/c nude mice	Tail vein injection, 20, 40, and 80 mg/kg	Upregulated HIPK2, p53 and Bax; downregulation Bcl-2	[357]

Abbreviation: ACC, acetyl CoA carboxylase; ACF, aberrant crypt foci; AFP, α-fetoprotein; AOM, azoxymethane; APC, adenomatous polyposis coli; BAX, B-cell lymphoma 2 associated x protein; BCAC, β-catenin accumulated crypts; BCL-2, B-cell lymphoma 2; BID, BH3 interacting-domain death agonist; CAC, colitis-associated colorectal cancer; CAT, catalase; CEA, carcinoembryonic antigen; CD, conjugated dienes; CIP2A, cancerous inhibitor of PP2A; c-MYC, cellular myelocytomatosis oncogene; COX-2, cyclooxygenase-2; CRC, colorectal cancer; CSK, C-terminal Src kinase; DACF, dysplastic aberrant crypt foci; DMH, 1,2-dimethylhydrazine; DNMT, DNA methyltransferase; EGCG, (-) epigallocatechin gallate; EGF-β, epidermal growth factor-β; EGFR, epidermal growth factor receptor; ERK, extracellular-signal-regulated kinase; FADD, Fas-associated protein with death domain; Flt-1, fms-like tyrosine kinase-1; GPx, glutathione peroxidase; GR, glutathione reductase; GSK-3β, glycogen synthase kinase-3β; GSH, glutathione; GST, glutathione S-transferase; γ-GT, γ-glutamyl transpeptidase; HMACF, high multiplicity aberrant crypt foci; IGF2, insulin like growth factor 2; IGFBP3, insulin like growth factor binding protein 3; IL-6, interleukin 6; iNOS, inducible nitric oxide synthase; KRAS, Kirsten rat sarcoma viral oncogene homolog; LC3b, light chain 3B of microtubule-associated proteins 1A/1B; LDH, lactate dehydrogenase; LMW-PTP, low molecular weight protein tyro-sine phosphatase; MAM, methyl azoxymethane; MAPK, mitogen-activated protein kinase; MDF, mucin-depleted foci; MMP, matrix metalloproteinase; mTOR, mammalian target of rapamycin; 5′-ND, 5′-nucleotidase; NEDD9, developmentally downregulated 9; NF-κβ, nuclear factor-κβ; Nrf-2, nuclear factor erythroid -2 related factor; ODC, ornithine decarboxylase; PCNA, proliferating cell nuclear antigen; PI3K, phosphoinositide 3-kinase; PP2A, protein phosphatase 2A; PTEN, phosphatase and TENsin homolog deleted on chromosome 10; QR, quinone reductase; SCID, severe combined immunodeficient; SIRT, Sirtuin 1; SOD, superoxide dismutase; STAT3, signal transducer and activator of transcription 3; TNF-α, tumor necrosis factor-α; Top1, topoisomerase 1; VDI, vascular density index; VEGF, vascular endothelial growth factor; γ-GT, γ-glutamyl transpeptidase.

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
