# Peer review of "Phenolic Phytochemicals for Prevention and Treatment of Colorectal Cancer: A Critical Evaluation of In Vivo Studies"

_cancers, 2023, doi:10.3390/cancers15030993_

Round 1

Reviewer 1 Report

I had the pleasure of reviewing the article, "Phenolic Phytochemicals for Prevention and Treatment of Colorectal Cancer: A Critical Evaluation of In Vivo Studies" by De et al., The authors should be commended for the extensive literature summary on this topic. The article is well-written, but I have a few comments/suggestions:

- The authors discussed the risk factors, genetic syndromes, chemo-prevention, and treatment at greater depth, which may not be needed for this article as the focus is primarily on phytochemicals. As such, would suggest considering trimming sections 1-5.

- It would be great if the authors discuss the clinical implications of the compounds discussed in the manuscript as it appeals the readership of Cancers.

- Would recommend having a section on future directions that summarize any available clinical trials using these compounds or if authors have a strong opinion on any of these compounds that have a "clinically meaningful" potential for clinical trial evaluation. 

Author Response

The authors of this manuscript express their sincere thanks to the reviewer for the critical assessment of this work. The authors have acted upon the recommendations of the reviewer which have resulted in a significant enhancement in the quality of this manuscript. All modifications incorporated in the manuscript are highlighted in a red color font. A “point-by-point” response to each comment is outlined below.

General comments:

I had the pleasure of reviewing the article, "Phenolic Phytochemicals for Prevention and Treatment of Colorectal Cancer: A Critical Evaluation of In Vivo Studies" by De et al., The authors should be commended for the extensive literature summary on this topic. The article is well-written, but I have a few comments/suggestions.

Response:

We would like to thank erudite reviewer for his/her appreciation of the manuscript. We have tried our best to address the comments raised by the reviewer and revised our manuscript.

Specific comments:

Comment 1:

The authors discussed the risk factors, genetic syndromes, chemo-prevention, and treatment at greater depth, which may not be needed for this article as the focus is primarily on phytochemicals. As such, would suggest considering trimming sections 1-5.

Response:

This is an excellent suggestion. We have significantly reduced the text under section 1 (page 2, line 59 to page 3, line 117) and section 3 (page 5, line 183 to page 6, line 229).

Comment 2:

It would be great if the authors discuss the clinical implications of the compounds discussed in the manuscript as it appeals the readership of Cancers.

Response:

We are in an absolute agreement with the reviewer. Accordingly, we have added additional information on clinical implication of various phenolic compounds in a new section “Phenolics in Clinical trials for CRC Treatment” (page 42, line 921 to page 43, line 942).

Comment 3:

Would recommend having a section on future directions that summarize any available clinical trials using these compounds or if authors have a strong opinion on any of these compounds that have a "clinically meaningful" potential for clinical trial evaluation. 

Response:

As mentioned above, we have added text to highlight various phenolic compounds which showed promising results in clinical trials (page 42, line 921 to page 43, line 942). While clinical trial results are encouraging, a major impediment to desirable success is the poor solubility and bioavailability of the polyphenols. Recent data suggested the most promising clinical potential with resveratrol. A similar potential is expected if the limitation due to curcumin’s poor solubility and bioavailability can be overcome with the synthesis of a suitable derivative. 

Additionally,

  1. The entire manuscript has been thoroughly checked and edited to minimize typographical errors as well as to ensure uniform style, organization, and quality.
  2. The reference list has been modified as we have deleted several new references. Special attention is given to conform to the order of references and bibliographic style of the journal.

Finally,

On behalf of my co-authors, I once again express my sincere thanks to the erudite reviewer for the valuable suggestions and constructive input to improve the quality of our manuscript.

Reviewer 2 Report

This submission represents a fairly comprehensive review of the impact of phenolic phytochemicals on CRC.

The review starts out strong when looking at CRC and risks, pathogenesis , prevention and treatment.  The review starts to get a little ragged when the main theme is addressed. This may be due to the scope of the coverage and the input of multiple authors. 

In the risk factors section, could we have more information on HR/RR for each of the risk factors, particularly since the issue of diet is raised?  Lines 178-188.

Minor-in Figure 1, box for article Eligibilit(y) and articled excluded we(a)k

Line 381-fewer side effects-compared to what? And references?

Phenolic compounds figures.  What divided the inclusion or exclusion of chemical structures.  Many of the ones discussed in depth or not in the figures. 

As a prelude to the discussion on separate compounds, it should be noted that some of the studies are using mixtures of compounds by something like necessity of sources as opposed to deliberate choice.  I suggest that where a deliberate choice was made to combine compounds that these studies be grouped into a separate section.  This does not include the Genistein and FOLFOX study.  I would recommend that where there are uses of pure or nearly pure compounds in studies that these be discussed first in each section and less pure compounds be listed later.  Be sure to make mention of which cell line was used in every section-for example lines, 409-412.

Also, when animal studies are mentioned, please be clear about the route of application and were possible whether the doses seem reasonable, or excessive-you do mention that some doses are below known toxic levels-can more information be supplied across more studies?  Did any seem to cross the threshold of safe doses?

Baicalin- Line 404- line 405 should probably be part of the paragraph above, otherwise line 404 appears to be an incomplete thought without a reference attached.

Section 7.7 Ge(I)nstein  spelling

Line 617-which non-enzymatic antioxidants?

Line 662 and 663-not sure what the response rate refers to?

The summary, lines 926-941-please expand by relating pathways for phytochemicals vs. Table 2.

Also, because of the seeming uniformly positive effects of the compounds, are there any unifying features-e.g. in oral administration for AOM induced tumors and impact on microbes that could be in common across multiple compounds?

When you mention the use of mixtures do you also include this kind of ridiculous current trend for  decoctions with so many components as to make any comment on mechanism impossible and sometimes seem to exist for the purpose of patentability or trademarking by institutions?

Author Response

The authors of this manuscript express their sincere thanks to the reviewer for the critical assessment of this work. The authors have acted upon the recommendations of the reviewer which have resulted in a significant enhancement in the quality of this manuscript. All modifications incorporated in the manuscript are highlighted in a red color font. A “point-by-point” response to each comment is outlined below.

General comments:

This submission represents a fairly comprehensive review of the impact of phenolic phytochemicals on CRC.

The review starts out strong when looking at CRC and risks, pathogenesis, prevention and treatment.  The review starts to get a little ragged when the main theme is addressed. This may be due to the scope of the coverage and the input of multiple authors.

Response:

We would like to thank erudite reviewer for his/her appreciation of the manuscript. We have tried our best to address the issue regarding the core area in our revised manuscript.

Specific comments:

Comment 1:

In the risk factors section, could we have more information on HR/RR for each of the risk factors, particularly since the issue of diet is raised?  Lines 178-188.

Response:

As per the reviewer’s suggestion, more information has been added (page 5, lines 159-182).

Comment 2:

Minor-in Figure 1, box for article Eligibilit(y) and articled excluded we(a)k

Response:

The word eligibility was placed in the box. The spelling “week” has been corrected to “weak” (page 10, line 330).

Comment 3:      

Line 381-fewer side effects-compared to what? And references?

Response

We have revised the sentence to indicate that the primary reasons for popularity of phytochemicals  include fewer side effects, easy availability, and low cost compared to the synthetic drugs with appropriate references (page 11, lines 347-349).

Comment 4:

Phenolic compounds figures.  What divided the inclusion or exclusion of chemical structures.  Many of the ones discussed in depth or not in the figures.

Response:

We sincerely apologize for inadvertent omission of several structures. We have provided new structure figures (Figures 2-5) to remedy the situation.

Comment 5:

As a prelude to the discussion on separate compounds, it should be noted that some of the studies are using mixtures of compounds by something like necessity of sources as opposed to deliberate choice.  I suggest that where a deliberate choice was made to combine compounds that these studies be grouped into a separate section.  This does not include the Genistein and FOLFOX study.  I would recommend that where there are uses of pure or nearly pure compounds in studies that these be discussed first in each section and less pure compounds be listed later.  Be sure to make mention of which cell line was used in every section-for example lines, 409-412.

Response:

We agree with the reviewer’s excellent suggestions. Accordingly, the discussion has been modified by indicating the cell lines as appropriate. For example, anti-CRC function of EGCG and acetylated EGCG was discussed (page 34, lines 504-521 ).  Moreover, the effect of EGCG in combination with other phytochemicals, such as curcumin was discussed in the revised manuscript (page 35, lines 522-529). The appropriate cell line has been mentioned while discussing the activity of a compound in context such as the in vivo antitumor activity of baicalin on HT-29 induced tumor xenograft developed in mice (page 32, line 424).

Comment 6:

Also, when animal studies are mentioned, please be clear about the route of application and were possible whether the doses seem reasonable or excessive-you do mention that some doses are below known toxic levels-can more information be supplied across more studies?  Did any seem to cross the threshold of safe doses?

Response:

We have added the route of administration of each of the phytochemicals along with their doses in the column 3 of Table 3 (pages 11-27). We also appreciate the reviewer’s insightful suggestion to include information on the toxicity/threshold level doses used in these studies. The dose used in these studies are not toxic. We mentioned the importance of this aspect in the conclusion section (page 45, lines 1013-1015).

Comment 7:

Baicalin- Line 404- line 405 should probably be part of the paragraph above, otherwise line 404 appears to be an incomplete thought without a reference attached.

Response:

We appreciate the reviewer’s close observation. The paragraphs were merged to improve the flow of information (page 33, lines 420-423).

Comment 8:

Section 7.7 Ge(I)nstein spelling

Response:

We apologize for this inadvertent error which has been rectified (page 35, line 567).

Comment 9:

Line 617-which non-enzymatic antioxidants?

Response:

We have added the names of enzymatic (SOD, CAT, GPx, and GR) and the nonenzymatic (vitamin E, vitamin C, vitamin A, and GSH) antioxidants which were measured in the study (page 35, lines 571-573).

Comment 10:

Line 662 and 663-not sure what the response rate refers to?

Response:

The study’s primary and secondary response rates were to understand the cycle 6 response rate, best overall response rate (BOR), and median progression-free survival (PFS) compared with the group not supplemented with genistein.  This has been explained in our revised manuscript (page 36, lines 617-619.

Comment 11:

The summary, lines 926-941-please expand by relating pathways for phytochemicals vs. Table 2.

Response:

We admire the reviewer for this excellent suggestion. We have described the group of phenolic compounds that exert their functions through inhibiting COX-2, or by inhibiting TPA-induced activation of AP1 of NF-kB (page 43, lines 969-973). We also highlighted phenolic compounds known for their antioxidant properties, or inhibitory action on mTOR or EGFR functions (page 43, lines 974-977).

Comment 12:

Also, because of the seeming uniformly positive effects of the compounds, are there any unifying features-e.g. in oral administration for AOM-induced tumors and impact on microbes that could be in common across multiple compounds?

Response:

We appreciate the reviewer’s suggestion to include the effect of phenolic compounds on the microbiome. We have included several studies that discussed the role of diet rich in phytochemicals in modulating population of microbiota implicated in maintaining gut homeostasis and reducing the risk of CRC incidence and/or progression in the conclusion and future perspective (Section 9: page 44, line 985 to page 45, line 996). We added a study of ameliorative effect of curcumin through maintenance of gut microbiota in AOM-treated IL10+/- mice (page 34, lines 470-474).

Comment 13:

When you mention the use of mixtures do you also include this kind of ridiculous current trend for  decoctions with so many components as to make any comment on mechanism impossible and sometimes seem to exist for the purpose of patentability or trademarking by institutions?

Response:

We understand the reviewer’s concern. We tried to avoid those studies that used a nonstoichiometric mixture of compounds and mainly focused on pure phenolic compounds.

Additionally,

  1. The entire manuscript has been thoroughly checked and edited to minimize typographical errors as well as to ensure uniform style, organization, and quality.
  2. The reference list has been modified as we have deleted several new references. Special attention is given to conform to the order of references and bibliographic style of the journal.

Finally,

On behalf of my co-authors, I once again express my sincere thanks to the erudite reviewer for the valuable suggestions and constructive input to improve the quality of our manuscript.

Round 2

Reviewer 2 Report

The paper looks good.  Just one minor spelling issue.  Rout(e) in a couple places in Table 2, towards end.

Author Response

We appreciate the reviewer's time to read our revised manuscript. We sincerely apologize for the inadvertent errors. We have corrected the minor issues with Table 3.